# Fluorogenic CRISPR for genomic DNA imaging

Zhongxuan Zhang[1,2,3,4,10], Xiaoxiao Rong [1,5,10], Tianjin Xie[1,6], Zehao Li[1,5], Haozhi Song[1], Shujun Zhen[6], Haifeng Wang[7], Jiahui Wu [8], Samie R. Jaffrey [9] & Xing Li [1,2,3,4] ✉

Genomic DNA exhibits high heterogeneity in terms of its dynamic within the nucleus, its structure and functional roles. CRISPR-based imaging approaches can image genomic loci in living cells. However, conventional CRISPR-based tools involve expressing constitutively fluorescent proteins, resulting in high background and nonspecific nucleolar signal. Here, we construct fluorogenic CRISPR (fCRISPR) to overcome these issues. fCRISPR is designed with dCas9, an engineered sgRNA, and a fluorogenic protein. Fluorogenic proteins are degraded unless they are bound to specific RNA hairpins. These hairpins are inserted into sgRNA, resulting in dCas9: sgRNA: fluorogenic protein ternary complexes that enable fluorogenic DNA imaging. With fCRISPR, we image various genomic DNA in different human cells with high signal-to-noise ratio and sensitivity. Furthermore, fCRISPR tracks chromosomes dynamics and length. fCRISPR also allows DNA double-strand breaks (DSBs) and repair to be tracked in real time. Taken together, fCRISPR offers a high-contrast and sensitive platform for imaging genomic loci.

Human genomic DNA and chromosomal functions are associated with unique subnuclear localizations that exhibit temporal dynamics. Effective imaging tools are needed to uncover this spatiotemporal information[1]. Various fluorescent tools, such as LacO and DNA-binding protein-fused fluorescent proteins, are broadly used to observe cellular chromosomes over time[2]. However, these tools require the insertion of exogenous sequences in the genome, thus requiring genome engineering as well as the possible interference in the function and dynamics of the chromosome from the exogenous sequences[2,3].

To image endogenous chromosomes, CRISPR (Clustered Regularly Interspaced Short Palindromic Repeats)/Cas9-based imaging tools have been developed. CRISPR/Cas9-based technologies are widely used for genomic DNA editing due to the ability of Cas9 to be targeted to specific genomic loci with high accuracy[4,5]. This feature was used to repurpose CRISPR to image endogenous genomic loci in living cells[2]. These imaging reporters are mainly catalytically deactivated Cas9 nuclease (dCas9)-fused fluorescent proteins or single guide RNA (sgRNA)-recruiting fluorescent protein-fused RNA-binding proteins[2,6–9]. However, these fluorescent proteins are constitutively fluorescent. Fluorescent proteins that are not bound to the genomic loci are found throughout the nucleus and have to be distinguished from the fluorescent proteins that are specifically bound to the genomic loci. The excess unbound fluorescent protein creates background fluorescence. Thus, this method has a low signal-to-noise ratio (SNR) and low sensitivity. Moreover, these constitutively fluorescent

[1]Beijing Institute of Life Sciences, Chinese Academy of Science, 100101 Beijing, China. [2]Department of Respiratory and Critical Care Medicine, The Affiliated Hospital of Southwest Medical University, 646000 Luzhou, Sichuan, China. [3]University of Chinese Academy of Sciences, 100049 Beijing, China. [4]State Key Laboratory of Integrated Management of Pest Insects and Rodents, Institute of Zoology, Chinese Academy of Sciences, 100101 Beijing, China. [5]College of Life Science, Hebei University, Baoding, 071002 Hebei, China. [6]School of Chemistry and Chemical Engineering, Southwest University, Beibei District 400715 Chongqing, China. [7]School of Life Sciences, Tsinghua-Peking Center for Life Sciences, Center for Synthetic and Systems Biology, Tsinghua University, 100084 Beijing, China. [8]Department of Chemistry, University of Massachusetts, Amherst, MA 01003, USA. [9]Department of Pharmacology, Weill Cornell Medicine, Cornell University, New York, NY 10065, USA. [10]These authors contributed equally: Zhongxuan Zhang, Xiaoxiao Rong. ✉e-mail: lix@biols.ac.cn

proteins often accumulate in the nucleolus and produce nonspecific nucleolar signals[2,3].

To reduce background fluorescence and increase fluorescence specifically associated with the genomic loci of interest, CRISPR was combined with fluorogenic RNA aptamers for genomic loci imaging[6,10]. Fluorogenic RNA aptamers can bind and fluorescently activate the nonfluorescent dyes[11–19]. However, few fluorogenic aptamers have the brightness needed for sensitive detection of genomic loci. Additionally, unlike fluorescent proteins, these dyes require exogenous addition as they are not genetically encoded. Therefore, the development of a genetically encoded and fluorogenic CRISPR-based system for efficient genomic DNA imaging would be highly desirable.

Here we describe a genetically encoded fluorogenic CRISPR (fCRISPR)-based system for genomic DNA imaging with low background fluorescence and high fluorescent activation. This approach uses a recently developed fluorogenic protein technology[20]. Fluorogenic proteins are highly unstable due to a Tat peptide-derived degron domain (tDeg), which causes rapid degradation unless the fluorogenic protein binds the Pepper RNA aptamer, which binds and conceals the degron. As a result, fluorescence is largely confined to the fluorogenic proteins that are bound to the Pepper RNA. We rationally modified the sgRNA with the Pepper hairpin to selectively localize fluorescence to the guide RNA while reducing fluorescence from unbound fluorogenic protein in the nucleus. The dCas9-sgRNA complex binds the target genomic DNA and stabilizes the fluorogenic protein for markedly enhanced genomic loci imaging. Overall, this CRISPR-based imaging system provides a robust platform for imaging DNA with minimal background fluorescence.

## Results

### Design of fCRISPR for genomic DNA imaging with low background fluorescence and high fluorogenic ability

To create a CRISPR-based genomic DNA loci imaging system with low background fluorescence and high fluorogenic ability, we chose Pepper-stabilized fluorogenic proteins as the fluorogenic reporter[20]. Unlike constitutively fluorescent proteins, fluorogenic proteins comprise fluorescent proteins fused with a Tat peptide-derived Degron domain (tDeg). Fluorogenic proteins are rapidly degraded by the cellular proteasome machinery but become stabilized and fluorescent when they bind to the Pepper aptamer (Fig. 1a)[20]. We confirmed that tDeg-fused red fluorescent protein tandem dimer Tomato (tdTomato-tDeg) was rapidly degraded by the proteasome machinery in living HEK293T cells. When Pepper RNA is expressed, tdTomato-tDeg is stabilized, exhibiting strong fluorescence in living cells (Supplementary Fig. 1).

We next sought to engineer a CRISPR-dCas9 imaging system using Pepper. sgRNAs in the CRISPR-dCas9 system have previously been adapted to recruit fluorescent proteins for imaging by fusing RNA hairpins (e.g., MS2, or PP7)[6,7]. Here, we inserted Pepper into both the tetraloop and stem-loop2 of the sgRNA scaffold (Fig. 1a and Supplementary Fig. 2a). In addition, fluorogenic proteins were expressed in order to assess their ability to bind and be stabilized by the Pepper-fused sgRNA.

We first asked if Pepper-fused sgRNA forms a functional complex with dCas9 for genomic imaging. To do this, we co-transfected three plasmids expressing (1) Pepper-fused sgRNA targeting the region of Chromosome 3[7,21]; (2) dCas9-fused green fluorescent protein (dCas9-GFP); and (3) the tdTomato-tDeg reporter, each into human bone osteosarcoma U2OS cells (Fig. 1b). A dCas9-fused fluorescent protein reporter has been reported for imaging genomic loci previously[2], and therefore was used as the positive control. To determine whether tdTomato-tDeg can be recruited to dCas9-GFP, we measured their fluorescence in cells. We found that the red fluorescent puncta from tdTomato-tDeg co-localized with the dCas9-GFP green fluorescent puncta (Fig. 1b), and this effect only occurred when the sgRNA was expressed (Supplementary Fig. 3a, b). These data suggest that a

ternary complex of tdTomato-tDeg:sgRNA:dCas9-GFP bound to genomic DNA is formed in cells.

We next asked if imaging genomic loci using fluorogenic proteins reduces background fluorescence and increases the imaging sensitivity compared to previous methods. The high background fluorescence limits the sensitivity and application of current CRISPR-based imaging tools. We found that the conventional genomic DNA loci imaging using the dCas9-GFP reporter showed high background fluorescence in both the nucleus and cytoplasm, reducing the SNR and imaging sensitivity (Fig. 1b). Moreover, the dCas9-GFP reporter showed nonspecific accumulation signal in nucleolus (Fig. 1c). Our results are consistent with previously reported results[2,3].

In contrast, CRISPR with Pepper: tdTomato-tDeg shows minimal background fluorescence compared to the dCas9-GFP reporter (Fig. 1b). In addition, unlike dCas9-GFP, which frequently accumulates in the nucleolus, the nonspecific accumulation signal was barely found using CRISPR with tdTomato-tDeg (Fig. 1c and Supplementary Fig. 4a). Also, the SNR of CRISPR using the Pepper system was ~26-fold higher than that of dCas9-GFP with a maximum of 116 SNR (Fig. 1d, e). Moreover, we compared two systems with the same tdTomato reporter, including CRISPR using Pepper: tdTomato-tDeg and the conventional CRISPR method[2] using dCas9-tdTomato. As expected, the SNR of CRISPR with Pepper: tdTomato-tDeg was significantly higher than that of dCas9-tdTomato (Supplementary Fig. 4). As CRISPR using the Pepper: fluorogenic protein system is highly fluorogenic, and exhibits low background fluorescence, we thus term it fluorogenic CRISPR (fCRISPR).

We next compared fCRISPR with the previously reported CRISPR system in which the guide RNA contains MS2 hairpins to recruit MS2 coat protein (MCP)-fused fluorescent proteins[22]. To test this, we constructed a sgRNA containing two MS2 hairpins by inserting the MS2 RNA hairpins into the tetraloop and the stem-loop2 of the sgRNA as previously described[6]. Next, we co-transfected three plasmids that express MS2-fused sgRNA, dCas9, and MCP-fused GFP in U2OS cells. As with Pepper-fused sgRNA in the fCRISPR system, the MS2-fused sgRNA targeted Chromosome 3. Like dCas9-fused fluorescent proteins, MCP-fused fluorescent proteins are non-fluorogenic and therefore showed high background fluorescence and low SNR (Supplementary Fig. 6a). In addition, we also observed distinct nonspecific nucleolar accumulation (Supplementary Fig. 6b). However, fCRISPR showed low background fluorescence and clear fluorescent puncta without accumulation (Supplementary Fig. 6c). We observed one to three labeled Chromosome 3 foci in various U2OS cells (Supplementary Fig. 5), which matches previous reports[6,7]. The CRISPR system with MCP-fused fluorescent protein reporter showed notably higher background fluorescence and lower SNR compared to fCRISPR (Supplementary Fig. 6). These results are consistent with the constitutively fluorescent nature of the MCP-fused fluorescent proteins compared to the fluorogenic nature of the fluorescent proteins in fCRISPR.

One potential problem is that sgRNA that is unbound by dCas9 could also bind and stabilize the tdTomato-tDeg. However, small RNAs are usually unstable in cells when not bound to proteins[6,10,23]. We, therefore, asked if Pepper-fused sgRNA is stable when not bound to dCas9. To test this, we performed real-time quantitative PCR (RT–qPCR) analysis for Pepper-fused sgRNA (Supplementary Fig. 2b). We found nearly all Pepper-fused sgRNAs are degraded in the absence of dCas9 (Supplementary Fig. 2b), which is consistent with the previous finding that sgRNAs are very unstable and degraded by ribonuclease A without dCas9[10]. Therefore, Pepper-fused sgRNA alone is unstable, and cannot stabilize fluorogenic protein to produce the background fluorescence.

In contrast, the sgRNA levels increased ~3000 fold in cells in the presence of dCas9 (Supplementary Fig. 2b). In addition, the fluorogenic protein does not affect the sgRNA instability in the absence of dCas9 or the stability of dCas9 (Supplementary

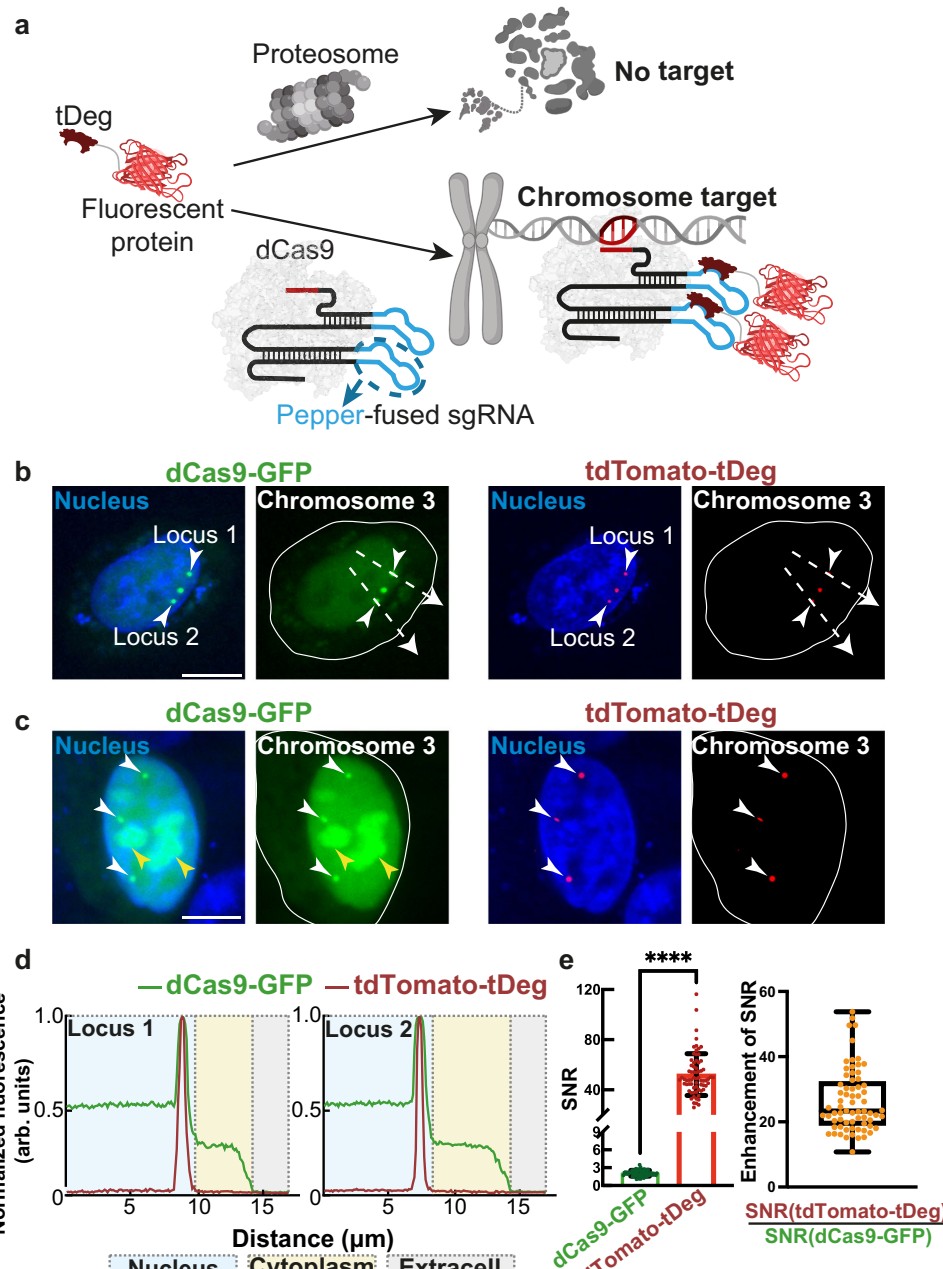

**Fig. 1 | Development of fCRISPR for genomic loci imaging with low background and high fluorescent activation. a** Schematic of fCRISPR with Pepper-stabilized fluorogenic proteins. We utilized a fluorescent protein (red) fused to a tDeg (dark red) as the fluorogenic protein. When fused to a fluorescent protein, tDeg causes this fusion protein to be degraded by the proteasome. However, tDeg destabilization is impeded when it binds to sgRNA containing Pepper (blue line) and spacer region (red line). Thus, Pepper-fused sgRNA forms a complex with dCas9 (light gray), enabling imaging of genomic DNAs with low background and high sensitivity. Figure 1a was created with BioRender.com. **b** Comparison of genomic loci labeling (white arrows) between conventional CRISPR with dCas9-GFP reporter and fCRISPR with tdTomato-tDeg reporter. The white dotted lines run through the nucleus to the extracellular environment to produce the plot-profile in Fig. 1d. **c** fCRISPR did not display the nonspecific accumulation fluorescence in the nucleolus (yellow arrow) and targeted Chromosome 3 (white arrow). **b, c** The white

circle line represents the whole cell. All cells were stained with Hoechst dye (1.0 μg/ml). Scale bar, 10 μm. These experiments were performed three times with similar results. **d** Fluorescence profiles of labeled Chromosome 3 loci (Loci 1, left; Loci 2, right) with dCas9-GFP reporter (green) and tdTomato-tDeg reporter (red). Fluorescence plots were generated from the white dotted lines in Fig. 1b. Source data are provided as a Source Data file. **e** fCRISPR (tdTomato-tDeg, red) shows a much higher SNR compared to the dCas9-GFP (green). For the bar plot (left), data are represented as means ± standard deviation for tdTomato-tDeg (52.14 ± 16.88) and dCas9-GFP (2.056 ± 0.476). ****$p = 3.4 \times 10^{-21}$ by Wilcoxon matched-pairs signed rank test was calculated with two-sided (left). For the box plot (right), the calculated ratios between tdTomato-tDeg and dCas9-GFP at each locus (orange) with 26.47 ± 9.66. The SD, mean, 5% and 95% percentiles are shown in the box plot (right). The whiskers extend to the minimum and maximum values (right). $n = 23$ cells with 69 fluorescent puncta. Source data are provided as a Source Data file.

Fig. 2b). These results indicate that fluorescent puncta from fluorogenic proteins that are stabilized by unbound sgRNA is negligible compared to fluorescence derived from fluorogenic proteins stabilized by dCas9-sgRNA complexes.

## fCRISPR enables various genomic loci imaging in living human cell lines

We next asked if fCRISPR can be used to image genomic loci in other cell lines. To test this, we expressed fCRISPR targeting Chromosome 3

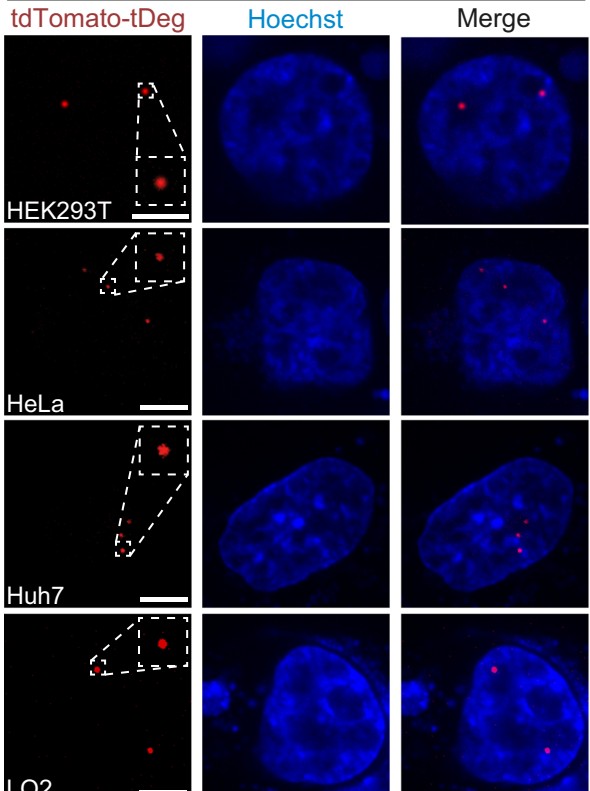

**a** **fCRISPR in various human cell lines**

tdTomato-tDeg | Hoechst | Merge

HEK293T

HeLa

Huh7

LO2

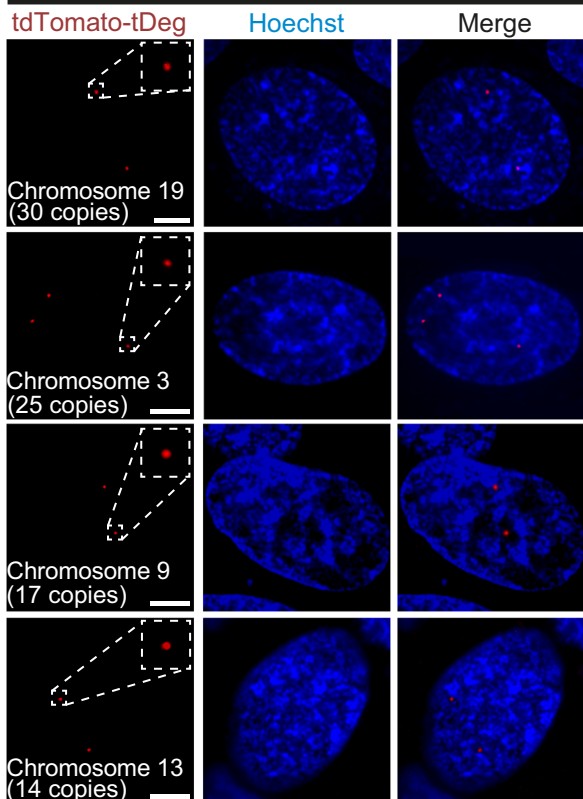

**b** **fCRISPR targets low-copy genomic loci**

tdTomato-tDeg | Hoechst | Merge

Chromosome 19 (30 copies)

Chromosome 3 (25 copies)

Chromosome 9 (17 copies)

Chromosome 13 (14 copies)

**Fig. 2 | fCRISPR-mediated imaging systems for various low-copy genomic loci tracking. a** fCRISPR enables genomic loci imaging in various living human cell lines. We showed that fCRISPR is able to image Chromosome 3 in living U2OS cells in Fig. 1. To test if fCRISPR is capable of imaging Chromosome 3 in other living human cells, we transfected fCRISPR with tdTomato-tDeg reporter in HEK293T, HeLa, Huh7, and LO2 cell lines, respectively. In all cell types observed, red fluorescent puncta from tdTomato-tDeg reporters were readily detected only when expressed using fCRISPR. These red fluorescent puncta co-localized with green fluorescent puncta from dCas9-GFP reporter in cells (Supplementary Fig. 7a). All cells were stained with Hoechst dye (1.0 μg/ml). Scale bar, 5 μm. *n* = 3 biologically independent experiments. **b** fCRISPR enables imaging of low-copy genomic loci in various

chromosomes. We showed that fCRISPR is able to image genomic loci with high-copy numbers in chromosomes in Fig. 1 and Supplementary Fig. 7. To test if fCRISPR is able to image low-copy genomic loci, we transfected fCRISPR with tdTomato-tDeg reporter toward low-copy genomic loci on Chromosome 19 (30 copies), Chromosome 3 (25 copies), Chromosome 9 (17 copies), and Chromosome 13 (14 copies), respectively, in U2OS cells. With fCRISPR, we readily detected all these low-copy genomic loci under the confocal microscope. These red fluorescent puncta co-localized with green fluorescent puncta from FITC-fused DNA FISH probes in cells (Supplementary Fig. 8b). All cells were stained with Hoechst dye (1.0 μg/ml). Scale bar, 5 μm. *n* = 3 biologically independent experiments.

in various human cell lines. We observed effective labeling of the target genomic loci with high SNR that co-localized with the fluorescent spots of dCas9-GFP in different human cell lines (Fig. 2a and Supplementary Fig. 7a). We observed different numbers of Chromosome 3 puncta in HEK293T, HeLa, Huh7, and LO2 cells, depending on the karyotype of each human cell line (Fig. 2a). Thus, fCRISPR can be used in various living human cell lines.

We next sought to target the diverse high-copy genomic loci on chromosomes. To do this, we transfected fCRISPR that targeted the repetitive region of either telomeres (>500 copies), centromeres (>500 copies), Chromosome 13 (~500 copies), or in intron 1 of the *MUC4* gene (*MUC4*-I1, ~90 copies) in U2OS cells (Supplementary Fig. 7b). Imaging showed the fluorescent puncta of tdTomato-tDeg reporter and dCas9-GFP reporter were co-localized substantially, while minimal tdTomato-tDeg fluorescence and no spots were seen in cells lacking sgRNA (Supplementary Fig. 3a). These results confirm that fCRISPR is able to target other genomic loci with high specificity.

We next asked if fCRISPR enables the detection of low-copy (~20 copies) genomic loci. To do this, we expressed fCRISPR that targeted various low-copy genomic loci ranging from 5 to 30 copies in Chromosomes 3, 9, 13, and 19 (Supplementary Table 3) in U2OS cells,

respectively. Imaging results showed that fCRISPR can label genomic loci with more than 14 copies (Fig. 2b and Supplementary Fig. 8b–d). In contrast, CRISPR with dCas9-GFP reporter failed to label genomic loci with copy numbers lower than 30, which is consistent with previous results[2,22] (Supplementary Fig. 8a). In addition, fCRISPR can detect the number of chromosomes by labeling low-copy genomic loci in U2OS cells (Supplementary Fig. 8e). These results demonstrate that fCRISPR can image low-copy genomic loci with high sensitivity.

**fCRISPR shows multi-color and orthogonal genomic loci imaging**

We next asked if fCRISPR is able to image genomic loci with various colors. Imaging color with fCRISPR is dependent on the fluorescent proteins fused on tDeg. We thus fused tDeg to the C-terminus of fluorescent proteins with different hues, resulting mCerulean-tDeg, mNeongreen-tDeg, YPet-tDeg, tdTomato-tDeg, and iRFP670-tDeg. These various tDeg-based reporters exhibit different emission wavelengths ranging from blue to near-infrared (NIR) (Fig. 3a). We next transfected fCRISPR with various tDeg-fused fluorescent proteins in living U2OS cells, respectively. All fluorogenic proteins showed effective labeling with high SNR (Fig. 3a and Supplementary Fig. 9). Thus, fCRISPR exhibits tunable fluorescence emissions.

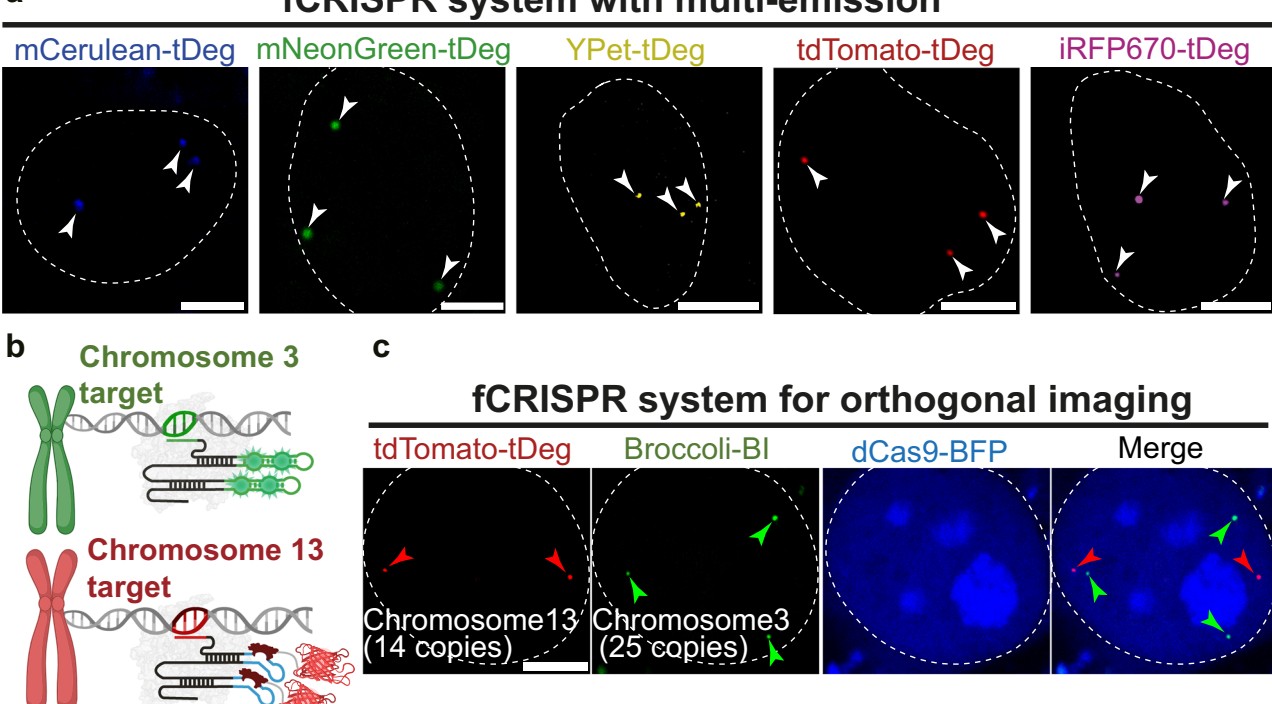

**Fig. 3 | Multi-color and orthogonal genomic loci imaging with fCRISPR.**
**a** fCRISPR enables multi-color imaging by modulating fluorescent proteins-tDeg. We showed that fCRISPR is able to image genomic loci with red fluorescence using tdTomato-tDeg reporter in Figs. 1 and 2. Here, we apply fCRISPR to image Chromosome 3 in living U2OS cells varying from blue to NIR color by fusing various colors of fluorescent proteins including mCerulean (blue), mNeonGreen (green), YPet (yellow), tdTomato (red), and iRFP670 (NIR) with tDeg. These results (white arrowheads) confirmed that fCRISPR can be developed to detect genomic loci with multiple colors. Scale bar, 5 μm. $n = 3$ biologically independent experiments.
**b** Schematic of dual genomic loci imaging with fCRISPR using tdTomato-tDeg reporter (lower figure), and CRISPR using Broccoli-BI reporter (upper figure). Both

CRISPR systems are orthogonal. Figure 3b was created with BioRender.com.
**c** Orthogonal imaging of low-copy genomic loci in Chromosome 3 and Chromosome 13 in U2OS cells. To simultaneously image low-copy genomic loci, such as Chromosome 3 (25 copies, green arrowheads) and Chromosome 13 (14 copies, red arrowheads), we co-expressed fCRISPR using tdTomato-tDeg reporter, and Broccoli-fused CRISPR using green fluorescent BI reporter. Both red and green fluorescent puncta were readily detected in the presence of BI dyes (10 μM) and co-localized with the blue fluorescent dCas9-BFP reporter in the merged image. These results demonstrated that fCRISPR can provide an orthogonal imaging platform with other CRISPR imaging systems in living cells. The white dotted circle line represents the nuclear. Scale bar, 5 μm. $n = 3$ biologically independent experiments.

We next asked if fCRISPR enables multiplexed genomic loci imaging with other CRISPR-based genomic loci imaging systems. To test this, we designed another CRISPR-based genomic imaging system. In this system, we fused the sgRNA with a fluorogenic aptamer, Broccoli (Supplementary Fig. 10a). Broccoli binds and activates the small molecule BI dyes[13–15,17] (Supplementary Fig. 10b). With Broccoli-fused CRISPR, we were able to detect different genomic loci with more than 20 copies (Supplementary Fig. 11a–c).

We next asked if fCRISPR and Broccoli-fused CRISPR can orthogonally detect two different genomic loci in the same cell. We transfected fCRISPR targeting genomic loci with 14 copies in Chromosome 13 using tdTomato-tDeg reporter and Broccoli-fused CRISPR targeting genomic loci with 25 copies in Chromosome 3 using the BI reporter in U2OS cells, simultaneously (Fig. 3c). We observed low-copy genomic loci in Chromosome 13 and Chromosome 3 spots in the same cells in the presence of BI (10 μM). In addition, we imaged two different high-copy (~500 copies) genomic loci with fCRISPR and Broccoli-fused CRISPR (Supplementary Fig. 10c). Together, these results suggest that fCRISPR can be combined with Broccoli-fused CRISPR for orthogonal genomic DNA imaging with various copies.

**fCRISPR reveals chromosomal dynamics heterogeneity**
We wanted to track chromosomal dynamics in living cells. Visualization of chromosomal dynamics is important for understanding fundamental intranuclear processes, such as how chromosomes dynamically control genomic functions[2,24]. To track chromosomal

movement, we expressed fCRISPR targeting Chromosome 3 in U2OS cells and performed high-frequency (0.3 s per frame) confocal microscopy imaging. We observed three fluorescent puncta of the fCRISPR reporter in cells (Fig. 4a), consistent with the triploid genome in the U2OS cell line[6,21]. Additionally, we observed the movement speed of each Chromosome 3 fluorescent spot was different (Supplementary Movie 1). We performed single-particle trajectory tracking in these three targeted loci, respectively. Single-particle tracking revealed the confined diffusion of each Chromosome 3 loci. The trajectories of each locus displayed confined movements at ~10 s timescale. Additionally, we observed that locus 2 showed directional transport, which was not found in locus 1 or 3 (Fig. 4a).

We next sought to assess the heterogeneity of chromosomal dynamics in living cells. To do this, we described these confined movements by the mean-squared displacement (MSD) curves with the microscopic diffusion coefficient ($_{micro}$). The MSD analysis gives the same results as the trajectories, i.e., that the movements of Chromosome 3 are highly confined and heterogeneous (Fig. 4b).

In addition, we conducted a comparison between fCRISPR and previously reported CRISPR-based imaging systems for chromosomal dynamic studies. CRISPR with dCas9-GFP reporter[2] and MS2-fused CRISPR with MCP-GFP reporter[7] have been previously reported for chromosomal dynamics imaging. We then compared fCRISPR and these two systems for imaging Chromosome 3 dynamics. We found that fCRISPR performed on par with the conventional CRISPR imaging systems for chromosomal dynamics imaging (Supplementary Fig. 12

**a** **Tracking Chromosome 3 locus movement**

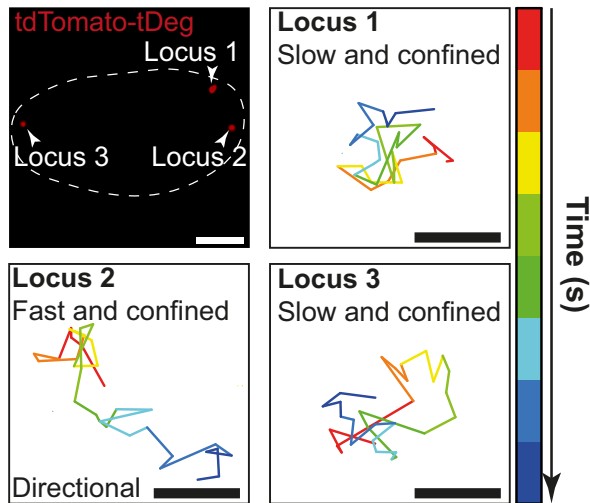

**b**

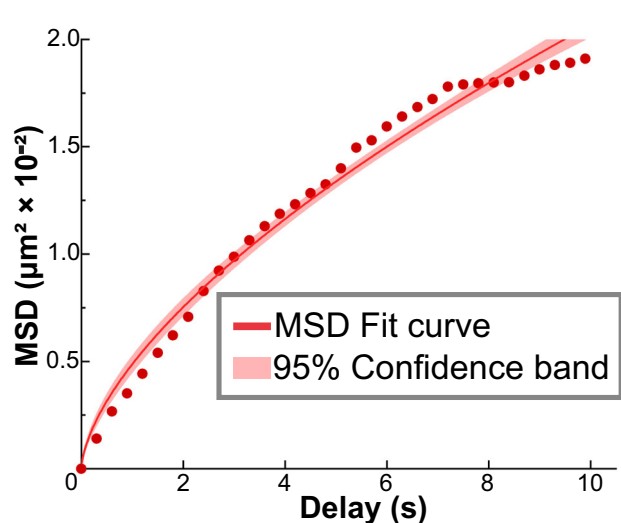

**Fig. 4 | fCRISPR reveals heterogeneity of chromosomal dynamics. a** fCRISPR reveals heterogeneity of chromosomal dynamics of Chromosome 3. To uncover the dynamics information of chromosome loci, we transfected fCRISPR with tdTomato-tDeg reporter targeting to Chromosome 3 in U2OS cells. We observed three fluorescent puncta (white arrowheads) and their movements in the nucleus (white dotted line). The results show that three Chromosome 3 loci have different movement speeds, revealing the heterogeneity of the specific genome loci. the time interval is 0.3 s, the gradient colors of each trajectory represent the from 0 s to

9.9 s. Scale bar for fluorescence figure, 5 μm. Scale bar for the trajectories, 0.2 μm. See Supplementary Movie 1. Source data are provided as a Source Data file. **b** Averaged MSD curves of fCRISPR-labeled Chromosome 3 in U2OS cells. The colored shaded area represents the 95% fitting confidence interval. The microscopic diffusion coefficient ($D_{micro}$) was $1.8 \times 10^{-3}$ μm²/s, which represents the confined movement of Chromosome 3. n = 30 cells were analyzed. Source data are provided as a Source Data file.

and Supplementary Movie 2–4). However, fCRISPR exhibits low background and high SNR, enabling it to readily track chromosomal dynamics in living cells.

### fCRISPR detects telomere length

We next asked whether fCRISPR can detect telomere length. Telomere length is associated with cellular lifespan and tumorigenesis[25]. Telomeres shorten gradually with each cell division and eventually shorten to a certain extent, leading to cell division arrest[26]. Thus, detecting telomere length in living cells allows real-time observation of telomere length changes and reveals the relationship between its changing and cellular state in real-time[2,9].

We next sought to detect telomeres in cells varying with telomere length. The telomere length in the human bladder cancer cell line (UMUC3) is shorter than that of retinal pigment epithelium (RPE) cell lines as cancer cell lines have a short telomere length[2,25]. We transfected dCas9-fused GFP, Pepper-fused sgRNA that targets telomere, and tdTomato-tDeg in RPE or UMUC3 cells. We observed the telomere fluorescent puncta under both tdTomato-tDeg reporter and dCas9-GFP reporter in the same cells. The telomere loci imaging with conventional CRISPR with dCas9-GFP[2] validated that fCRISPR with tdTomato-tDeg can image telomere (Supplementary Fig. 13). However, the background fluorescence of the tdTomato-tDeg reporter was significantly lower than the dCas9-GFP reporter (Supplementary Fig. 13).

We next compared the brightness of telomere puncta between RPE and UMUC3 cell lines. The results showed that the telomere length of RPE is 3.11-fold and 2.85-fold longer than UMUC3 when detected by fCRISPR and conventional CRISPR, respectively (Fig. 5), which is consistent with previous study[2]. Therefore, fCRISPR did not show an obvious difference compared to the conventional CRISPR imaging system for telomere length detection. Together, fCRISPR can detect the relative fluorescence brightness of telomeres, thereby inferring telomere length.

### fCRISPR tracks DNA double-strand breaks and repairs

We next asked whether fCRISPR can be used to track DNA breaks and repairs. DNA double-strand breaks (DSBs) are one of the most severe types of DNA damage that lead to the development of major diseases such as gene mutations, genomic instability, and tumor-related diseases. Thus, observing or monitoring the dynamics of the induced DNA breaks and repairs in real-time is vital to understanding the mechanism of DSBs in cells[3,21].

Thus, we sought to spatiotemporally image DNA breaks in cells. P53-binding protein 1 (53BP1) plays a key regulatory role in DSBs repair and is recruited to the DSBs site[3]. Thus, we created a U2OS cell line that stably expresses truncated 53BP1-fused to the Apple (53BP1-Apple) red fluorescent protein (Supplementary Fig. 14). We demonstrated that 53BP1-Apple is a sensor for DNA breaks by multiple validation approaches (Supplementary Fig. 14), which is consistent with the previous reports[21,27].

Next, to induce DNA breaks, we co-transfected the Cas9 protein with a *PPP1R2*-targeted sgRNA that induces DNA breaks in a ~ 36 kb domain within the repetitive region of Chromosome 3[21]. In addition, we co-transfected fCRISPR with YPet-tDeg to track genomic loci on Chromosome 3 (Fig. 6a). After 7 h transfection, we observed 53BP1-Apple and fCRISPR with YPet-tDeg gradually colocalizing (Fig. 6a). We confirmed the recruitment of multiple repair factors (53BP1, and γH2AX) to the single break sites (Supplementary Figs. 14 and 15). These data suggest Cas9-induced DSB formation at the *PPP1R2* locus and the ability of fCRISPR to label this locus in living cells (Supplementary Fig. 14).

After the recruitment of 53BP1 foci to Chromosome 3, we observed subsequent dissociation of 53BP1 foci to Chromosome 3 for hours (Fig. 6b, c; Supplementary Fig. 16a, Supplementary Movie 5). These data suggest that the dissociation of 53BP1 from Chromosome 3 likely represents DSBs that were successfully repaired[21]. The dwell time of 53BP1 was different, mostly varying in 2–4 h[27,28] (Supplementary Fig. 16b). These data suggest that the timing of repair after

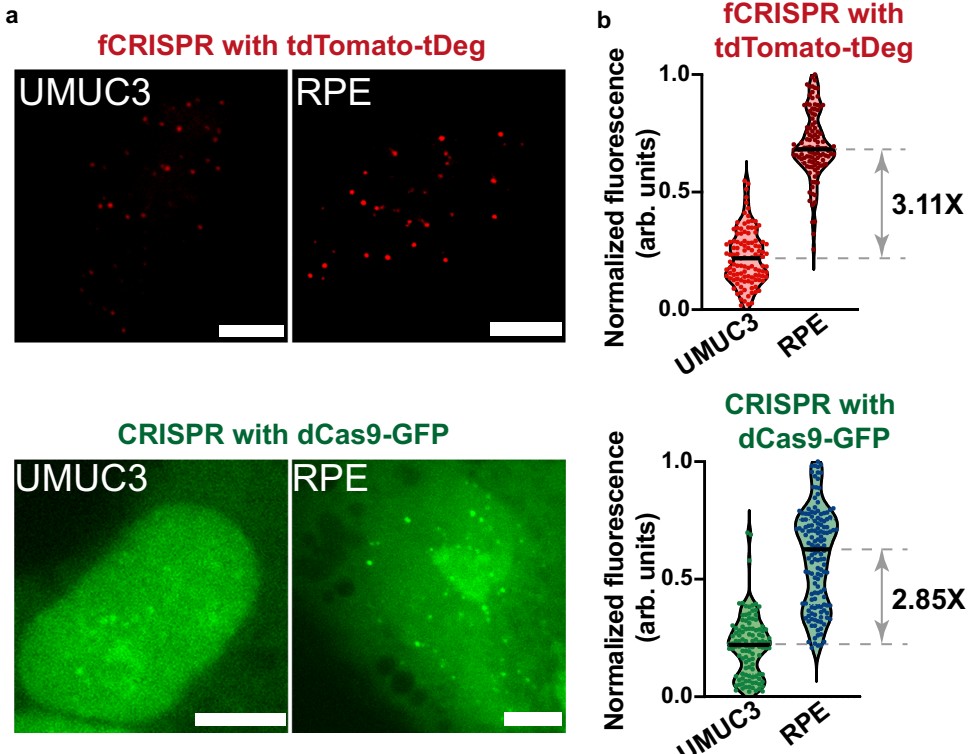

**Fig. 5 | fCRISPR detects the telomere length in UMUC3 and RPE cell lines.**
**a** UMUC3 and RPE cells expressing fCRISPR with tdTomato-tDeg or conventional CRISPR with dCas9-GFP were imaged to determine telomere length. To image telomere, we transfected fCRISPR (top) and conventional CRISPR (bottom) in living UMUC3 (left) and RPE (right) cells. Scale bar, 5 μm. **b** Quantification of normalized telomere fluorescence labeled by fCRISPR or conventional CRISPR in UMUC3 and RPE cells. The mean fluorescence of telomere puncta labeled by fCRISPR in RPE cells showed ~3.11-fold higher than in UMUC3 cells. The mean fluorescence of telomere puncta labeled by conventional CRISPR in RPE cells showed ~2.85-fold higher than in UMUC3 cells. The shown dots represent the mean fluorescence of single cells. $n = 400$ cells per condition. Source data are provided as a Source Data file.

Cas9-induced DSBs is heterogeneous[27]. Therefore, fCRISPR is capable of imaging DNA breaks and repairs events in living human cells.

We also observed the additional DSBs and their repair events. We found 53BP1-Apple foci repeatedly recruitment and resolution at *PPP1R2* loci on both homologous Chromosome 3, possibly indicating the repeated cutting and repairing of the target DNA[21,27] (Supplementary Fig. 17 and Supplementary Movie 6). In addition, we observed two 53BP1 foci that initially formed at separate Chromosome 3 loci, then stayed together, which likely suggests chromosomal interactions between the two DSBs loci (Supplementary Fig. 18 and Supplementary Movie 7)[21].

Furthermore, we performed fCRISPR to image DSB and repair events in other chromosomes in addition to Chromosome 3. We expressed fCRISPR to image Chromosome 13 and active CRISPR to induce DSBs at the *SPACA7* locus, which is ~82 kb away from the imaging locus (Supplementary Fig. 19a). As respected, we observed 53BP1-Apple recruitment and dissociation on Chromosome 13, which indicates that fCRISPR can track DSBs and repairs in different chromosomes (Supplementary Fig. 19 and Supplementary Movie 8).

## Discussion
Here, we report fCRISPR, a system for improved imaging of genomic loci in living cells. This system relies on the recruitment of a fluorogenic protein to Pepper inserted into the sgRNA. The fluorogenic proteins exhibit fluorescence only when bound to the dCas9-sgRNA complex that is targeted to genomic loci. In this way, background fluorescence that derives from unbound fluorescent protein is markedly reduced. Unlike the previous CRISPR-based imaging tools which exhibit constitutive fluorescence and high background fluorescence, our fCRISPR shows low background fluorescence and fluorogenic

ability, thus increasing the sensitivity of genomic DNA imaging. These features allow fCRISPR to function as an efficient, robust, and scalable genomic DNA imaging platform.

To create fCRISPR, we modified the sgRNA with Pepper, which stabilizes the otherwise rapidly degraded tdTomato-tDeg fusion protein. The dCas9 is guided to genomic loci via the Pepper-fused sgRNA, and the fluorescent puncta is induced by the sgRNA which stabilizes the fluorogenic protein. sgRNAs that are not bound to dCas9 are degraded, which reduces the nonspecific stabilization of fluorogenic protein. The degradation of fluorogenic protein is the basis for the fluorescent noise reduction in fCRISPR.

In addition, when compared to genomic imaging using dCas9-fused fluorescent proteins or MS2-modified sgRNA recruiting MCP-fluorescent proteins, the fluorescent proteins caused diffuse nucleoplasmic background fluorescence and were frequently found accumulated in the nucleolus[2,29]. However, fCRISPR notably exhibits a lower nucleoplasmic background and negligible nucleolar accumulation. These features are highly beneficial for imaging studies.

The low background and high SNR characteristics of fCRISPR enable us to detect low-copy (~20 copies) genomic loci without signal amplification. fCRISPR is designed with single sgRNA fusing with two short Pepper aptamers (28 nt)[20], recruiting two fluorogenic proteins. With fCRISPR, we are able to image genomic loci as low as 14 copies under a normal confocal microscope. However, other conventional systems are unlikely to detect such low-copy genomic loci without signal amplification[6,22,30]. Therefore, our fCRISPR system showed high labeling sensitivity and served as an important step toward genomic loci imaging at high spatiotemporal resolution.

Furthermore, we successfully imaged the dynamic movements of genomic loci in different human cell lines in real-time. Additionally,

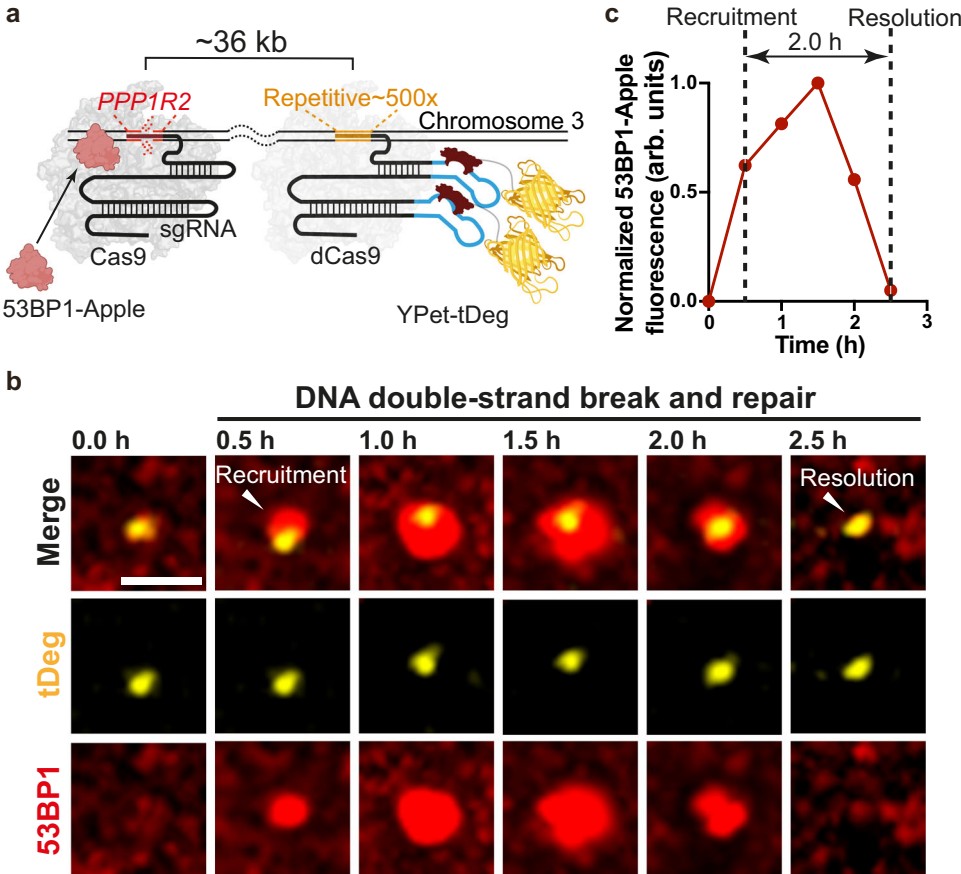

**Fig. 6 | fCRISPR tracking of DNA breaks and repairs in real-time. a** Schematic of fCRISPR with editing CRISPR for tracking of DNA breaks and repairs in living cells. We aimed to use fCRISPR to reveal the DNA breaks and repair mechanisms. To do this, we transfected the editing CRISPR-Cas9 (dark gray, crystal structure) to cut the *PPP1R2* (red) genomic loci (~36 kb from the repetitive region) of Chromosome 3 (orange). Once the *PPP1R2* genomic locus is broken, 53BP1 proteins (pink) are recruited at the *PPP1R2* genomic loci until the broken chromosomes are repaired. In addition, we transfected the fCRISPR with tDeg (dark red)-fused YPet (yellow) reporter for tracking the repetitive region of Chromosome 3 in U2OS cells. Figure 6a was created with BioRender.com. **b** Representative images of DNA breaks and repairs during the recruitment and resolution of 53BP1 at the *PPP1R2* locus. To image DNA breaks and repairs in living cells, we created U2OS cells that stably expressed a red fluorescent 53BP1-Apple reporter. Next, we transfected the Cas9 to cut the *PPP1R2* locus and transfected fCRISPR with a yellow fluorescent YPet-tDeg reporter to image Chromosome 3. These images showed that 53BP1 were recruited at the DNA breaks site at 0.5 h, and were resolved during DNA repairs at Chromosome 3 at 2.5 h. Images were acquired with the Olympus SpinSR-10 microscopy at 0.5 h time intervals. Z-stacks images were obtained at all time points. Image acquisition time, 2.5 h. The whole cell images and three-dimensional reconstruction are shown in Supplementary Fig. 16a and Supplementary Movie 5. Scale bar, 5 μm. The experiments were repeated three times independently with similar results. **c** Normalized fluorescence intensity of the 53BP1-Apple reporter during gene editing at the *PPP1R2* locus shown in Fig. 5b. Data were analyzed by Fiji, and processed by GraphPad Prism 9. Source data are provided as a Source Data file.

fCRISPR enabled tracking of an individual locus on different copies of the same chromosome in the nucleus, revealing dynamic heterogeneity between the chromosomes. We also combined an orthogonal CRISPR-based imaging system for the simultaneous imaging of two genomic loci in living cells.

fCRISPR also allows for dynamic tracking of Cas9-induced gene editing and DNA DSBs repair events at endogenous genomic loci in living human cells. Unlike ionizing radiation that induces DSBs in all cellular genomic DNA, Cas9 is able to induce DNA breaks at specific DNA loci with sgRNA. In addition, light-activated Cas9[31,32] or sgRNA[27] strategies enable study on DSBs and repair with temporal resolution. With fCRISPR, we thus spatiotemporally study the Cas9-induced DSB breaks and repair events, including DNA repair timing, for the desired DNA locus.

Like CRISPR LiveFISH[21], fCRISPR visualizes various DNA breaks and repairing events in living human cells. For example, fCRISPR and CRISPR LiveFISH observed the repeated cutting and repairing, as well as possible chromosome interaction[21]. However, since fCRISPR is genetically encoded, and does not need to label sgRNA with small molecule fluorophore by the laborious and complicated chemical modification[33]. Therefore, fCRISPR provides a robust and easy-to-operate approach for studying DNA breaks and repair events in living cells with high SNR and low background.

Like CRISPR with dCas9-fused fluorescent proteins, fCRISPR could visualize real unique sequences by designing a larger number of unique sgRNAs with different spacers[2]. However, this approach potentially exhibits several limitations. For example, this approach is difficult to implement for cellular applications due to the challenges in the delivery of a large number of sgRNAs into the same cells. Furthermore, this approach could increase the off-target effect by dozens of sgRNAs with various spacers[22]. Further study will be performed by designing one sgRNA fusing multiple fluorogenic proteins for fCRISPR signal amplification.

One potential problem is that the dCas9-sgRNA complex that is not bound to the genomic DNA can stabilize the fluorogenic protein, causing the background fluorescence. This is resolved by expressing dCas9 at low enough levels to match the expected number of dCas9:sgRNA binding sites on the chromosome.

Overall, our data showed that background fluorescence from fCRISPR is significantly lower compared to other fluorescent reporters

(e.g., dCas9-fused or MCP-fused fluorescent proteins) that are used in CRISPR-based imaging systems. The overall fluorogenic design principles described here may enable future improvement and optimization of CRISPR-based imaging tools.

## Methods

### Plasmids construction

To construct plasmids expressing dCas9-fused fluorescent proteins, we used pHAGE-TO-dCas9 (Addgene#75381) plasmids as the backbone. In this study, dCas9 was fused 2× SV40 nuclear localization sequence (NLS) and inserted fluorescent protein at the C-terminus by homologous directed recombination (Vazyme) methods. Fluorescent proteins included tagBFP (dCas9-BFP), super-fold GFP (dCas9-GFP), or 3× mCherry (dCas9-mCherry).

To construct plasmids expressing fluorogenic protein, we constructed a mini CMV promoter with bGH polyA (bovine growth hormone polyadenylation) signal for weak promotion and termination expression vector[20]. Various fluorescent proteins including tdTomato, YPet, mCerulean, mNeongreen, or iRFP670 were fused to the N-terminus of tDeg peptide by homologous directed recombination methods. In detail, the SV40 NLS was added at the N-terminus of the tDeg and C-terminus of the above fluorescent proteins except for mNeongreen. mNeongreen-tDeg was added dual SV40 NLS at the C and N-terminus of mNeongreen for sufficient nuclear localization.

To engineer plasmids encoding RNA-guided different loci with Pepper scaffold, we used gRNA_Cloning Vector (Addgene#41824) plasmid as the backbone. sgRNA contains G-C flip and hairpin extension (GGCC stem) to increase targeting efficiency and stability and is expressed under the human U6 promoter. To increase imaging capability purpose, Pepper sequences were added at both tetraloop and stem-loop 2. The spacer regions were changed by using site-directed mutagenesis (Vazyme) or restriction digested (New England Biolabs) and T4 DNA ligation methods (Vazyme). The spacer sequences are shown in Supplementary Table 3. All of the methods used for plasmid construction were following the manufacturer's recommended protocol.

### Cell culture

The U2OS (American Type Culture Collection (ATCC)-HTB-96), HeLa (ATCC-CCL-2), Huh7 (ATCC-CCL-185), LO2 (ATCC-HL-7702), HEK293T (ATCC-CRL-11268), RPE (ATCC-CRL-2302), and UMUC3 (ATCC-CRL-1749) cell lines were cultured in high glucose DMEM medium (Gibco) with 10% (vol/vol) Fetal Bovine Serum (Gibco), and 1% (vol/vol) Penicillin-Streptomycin (NCM Biotech) supplement. Cells were maintained at 37 °C and 5% CO2 humidified incubator. All cells were dissociated using 0.25% trypsin-EDTA (Macgene) according to the manufacturer's instructions. The stable transfer cell lines of U2OS expressed 53BP1-Apple were using the same culture and treatment conditions.

### Cell transfection

Cells were seeded at a density of 300,000 (U2OS, LO2) or 200,000 (HEK293T, Hela, and Huh7) cells/compartment in 12-well culture plates (Corning) 24 h before transfection. Cells were transfected with corresponding plasmids and used up to 2 μl Lipofectamine 3000 and p3000 transfection reagents (Thermo Fisher Scientific) in this study. The detailed transfection amounts are shown in Supplementary Table 1.

### Lentiviral production, transduction, and FACS

For lentiviral production, HEK293T cells were seeded at a density of 400,000 HEK293T cells/compartment (Corning) into 6-well culture plates 24 h before transfection. For transfection, 2 μg 53BP1-Apple plasmids, 1.5 μg of psPA × 2 plasmids, and 1 μg of pMD2.G plasmids. Meanwhile, added 10 μl P3000 and Opti-MEM up to 50 μl following the manufacturer's recommended protocol. Media was refreshed 6 h after transfection. Virus was collected 24 and 48 h after the first media refreshment. For viral transduction, U2OS cells were incubated with

virus solution in DMEM medium for 24 h. To make a stable cell line, cells were transduced with lentivirus, and selected with appropriate antibiotics (4–6 mg/ml puromycin or blasticidin) for 3 days. The medium containing dead cells was replaced every day.

Then the cells were sorted by fluorescence-activated cell sorting (FACS) at the Institute of Genetics and Developmental Biology in the Chinese Academy of Science using FACSAria III with 561 nm excitation. The transduced cells were sorted for cells that expressed low 53BP1-Apple to create stable cell lines. Cells were initially gated to FSC-A and SSC-A according to the observed singlets and doublets, selecting only singlets. Subsequently, the majority population showing a relatively lower 53BP1-Apple signal was selected for gating.

### RT−qPCR

Cells were transfected as described in the Cell transfection section. Briefly, 200 ng of dCas9 plasmid DNA, 300 ng of fluorescent protein-fused tDeg, and 500 ng of Pepper-fused sgRNA plasmids were co-transfected using Lipofectamine 3000 (Thermo Fisher Scientific), and the cells were incubated for another 48–72 h before collection. Whole cell RNA was extracted with trizol chloroform RNA extraction method. Briefly, cells were disrupted by adding 800 μl TRIzole reagent (Ambion) and 200 μl chloroform ($CHCl_3$). After centrifugation, RNA was extracted and purified by adding 500 μl Isopropanol with 1000 μl pre-cold ethanol. After centrifugation, the precipitate was collected and resuspended with RNase free water. cDNA was reverse transcribed by using PrimeScript RT reagent kit (Takara) with the following primers for Pepper-fused sgRNA: forward primer, 5′-TAGCAAGTTCAAAT AAGGCT-3′; reverse primer, 5′-GACTCGGTGCCACTTG-3′, and following the manufacturer's recommended protocol. RT−qPCR using the UltraSYBR One-Step RT−qPCR Kit (CWBIO) for the quantification of Pepper-fused sgRNA and β-actin mRNA. All data were collected with BioRad CFX manager and normalized for the cell number using β-actin mRNA as the internal reference using Excel software.

### Western blot (WB)

Briefly, 500 ng *PPP1R2*-targeted Cas9 plasmids DNA were transfected using Lipofectamine 3000 (Thermo Fisher Scientific), and the cells were incubated for another 7 h before collection. After 7 h transfection, cells were added to different concentrations of ATM inhibitor (KU-0055933) or DMSO, and incubated for 1 h. Then, cells were lysed and proteins were extracted using RIPA lysis and extraction buffer (Thermo Fisher Scientific) with Protease and Phosphatase Inhibitor cocktail (Thermo Fisher Scientific). The protein concentration is determined using a BCA quantification method with Pierce™ Rapid Gold BCA Protein Assay Kit (Thermo Fisher Scientific).

Subsequently, the proteins were separated by size using SDS-PAGE gel electrophoresis. The gel was then transferred onto a PVDF membrane (0.22 μm, Invitrogen) through a process known as blotting for 3 h. Then, the membrane was cut due to different primary antibody incubation needs. The membrane was blocked with a blocking buffer (5% nonfat dry milk dissolved in 1xTBST) to prevent nonspecific binding following transfer. The membrane was then cut for desired antibody incubation at 4 °C overnight. The phospho-ATM (Ser1981) primary Rabbit antibody, Phospho-Histone H2AX (Ser139) primary Rabbit antibody, and β-Actin (13E5) Rabbit antibody with 1:1000 dilution in 1% nonfat milk and purchased from Cell Signaling Technology (CST). After washing to remove unbound antibodies with 1xTBST, the membrane was incubated with a secondary Goat-anti Rabbit antibody conjugated to HRP (CST) that binds to the primary antibody at 4 °C for 4 h. Further washing was performed to remove unbound secondary antibodies with 1xTBST. Finally, the protein bands are detected using chemiluminescence, and the signals are recorded using the Tanon 5200 Chemiluminescent Imaging System and analyzed using FIJI.

## Live-cell imaging with fCRISPR

To image with confocal microscopy, cells were subcultured to 35 mm imaging dishes (Corning) at the desired time post-transfection. The imaging dishes for HEK293T were precoated with poly-ᴅ-lysine (Sigma-Aldrich) to maintain cell viability and normal growth characteristics. Before imaging, cells were changed to the imaging medium (Macgene) containing 1.0 µg/ml Hoechst 33342 (Thermo Fisher Scientific). One hour after staining, images were acquired with confocal microscopy (Olympus SpinSR10).

## Fluorescence in situ hybridization (FISH)

To validate the fCRISPR labeling chromosome loci, fCRISPR and FISH imaging were performed simultaneously. Cells were fixed with 4% paraformaldehyde for 15 min, then permeabilized with 0.7% Triton X-100, and 0.1% Saponin in 2× SSC for 30 min at room temperature. Cells were washed twice with 2xSSC, and treated with RNase A at 37 °C for 1 h, then rewashed with 2xSSC and equilibrated in PBS for 5 min. Cells were dehydrated through consecutive 5 min incubations in 70%, 85%, and 100% ethanol with air drying. After air drying, cells were incubated in 70% formamide/2xSSC for 5 min and washed using 70%, 85%, and 100% ethanol on ice. Cells were again air dried before the addition of Cy3/FITC-labeled oligo FISH probes (2 ng/ml final concentration, each oligo has one molecule of Cy3/FITC conjugated to its 5 prime end, IDT custom DNA oligo, sequences shown in Supplementary Table 3) in hybridizing solution (10% dextran sulfate, 50% formamide, 500 ng/ml salmon sperm DNA in 2×SSC). Cells were incubated overnight at 37 °C shielded from light. After hybridization, cells were washed with 2xSSC three times and finally mounted with ProLong® Gold Antifade Reagent with DAPI (CST). One hour after mounting, images were acquired with confocal microscopy (Olympus SpinSR10).

## Immunofluorescence (IF)

The desired cells were fixed in 4% paraformaldehyde for 15 min. After 1xPBS was washed, 0.1% Triton X-100 was added for 4 min on ice for permeabilization. Cells were washed with 1xPBS three times after permeabilization, then incubated with γH2AX (Phospho-Histone H2AX (Ser139) (20E3) Rabbit mAb #9718, CST) with 1:200 dilution in 1xTBST at 4 °C overnight. Then, the cells were pre-warmed at room temperature for 15 min, and washed with 1xTBST four times. Cells were shaded and incubated with Alexa Fluor® 488 conjugated secondary Goat-anti Rabbit antibody (CST) at room temperature for 2 h. Then, cells were washed four times using 1xTBST. After secondary antibody incubation, cells were finally mounted with ProLong® Gold Antifade Reagent with DAPI (CST). One hour after mounting, images were acquired with confocal microscopy (Olympus SpinSR10).

## Microscopy (Olympus SpinSR10)

Conditions were maintained at 37 °C and 5% CO$_2$. Live-cell imaging was conducted using a Photometrics Prime BSI camera with a confocal microscope (Olympus SpinSR10). The cellSens Dimension software (Olympus) was used to operate the microscope and camera. The conditions during live-cell imaging were maintained at 37 °C and 5% CO$_2$. Cells were imaged with UPLSAPO20X or UPLXAPO100XO objective. Hoechst dye and BFP were detected using a BFP filter cube (excitation filter 405 nm; emission filter 447 ± 30 nm) with an exposure time of 500 ms. mNeonGreen and GFP were imaged using a FITC filter cube (excitation filter 488 nm; emission filter 525 ± 25 nm) with an exposure time of 500 ms. YPet was imaged using a YFP filter cube (excitation filter 514 nm; emission filter 545 ± 20 nm) with an exposure time of 500 ms. Cy3 fluorophores, mCherry, mApple, and tdTomato were imaged using an RFP filter cube (excitation filter 561 nm; emission filter 617 ± 36.5 nm) with an exposure time of 500 ms. iRFP670 was imaged using a NIR filter cube (excitation filter 640 nm; emission filter 685 ± 20 nm) with an exposure time of 500 ms.

## SNR quantification

SNR is defined as the ratio of the intensity of a fluorescent puncta and the power of background fluorescence, and the straightforward steps of SNR calculation are shown below. The SNR is calculated with the following formula: SNR = $P_{signal}/P_{background}$ = Max intensity of spots signal/(Std. dev. of background – Std. dev. of extracellular background). All the fluorescence imaging data were analyzed by the Analyze Particles function in Fiji (version 2.9.0).

## Plot-profile quantification and analysis

The original data of plot profiles were exported from Fiji. We normalized the exported data and analyzed and created the figure by Origin. See figure legends for details.

## Statistical analysis of SNR and RT−qPCR

Data were exported from Fiji and analyzed by GraphPad Prism 9. For the comparisons, the Unpaired Student's t-test (Wilcoxon test) and correlation analysis were used in this study. $p < 0.05$ was considered significant and $p > 0.05$ was considered not significant. Error bars represent standard deviation (SD) from data in at least triplicate experiments and these are stated in the corresponding legends. See figure legends for details.

## Fluorescence imaging data colocalization and Z-stack processing

Colocalization analysis of fluorescent puncta labeled by dCas9, tDeg, or MCP-fused fluorescent protein in live-cell imaging was processed by Merge function in Fiji. The Stack function in Fiji processed the Z-stack images at maximum intensity.

## Single-particle tracking analysis

Chromosome 3 image stacks were first detected by TrackMate plugins in Fiji and analyzed by MATLAB tracking package 'msdanalyzer'[34]. Fluorescent puncta were identified in each frame by using LoG (Laplacian of Gaussian) detector with a suitable estimated object diameter and quality threshold. Then we used a simple LAP tracker for tracking. The nearest neighbors identified puncta within a maximum distance range of 5 camera pixels in the previous frame. The gap larger than 7 consecutive frames was treated as two particles. The data were exported from Fiji.

For each trajectory, the data were exported from 'Confined motion trajectories' function of 'msdanalyzer' and used Origin to add time coordinates shown by gradient colors. The MSD as a function of time delay was calculated by the following equation[2]:

$$MSD(n\delta t) = \frac{1}{N-1-n} \sum_{j=1}^{N-1-n} \left\{ \left[ x(j\delta t + n\delta t) - x(j\delta t)^2 \right] + \left[ y(j\delta t + n\delta t) - x(j\delta t)^2 \right] \right\} \tag{1}$$

where δt is the time interval between two successive frames, x(t) and y(t) are the coordinates at time t, N is the total number of frames and n is the number of time intervals. The analysis of MSD curves used the 'Fit of the linear part of the MSD' function of 'msdanalyzer' through log-log representation for confined diffusion, macroscopic or microscopic diffusion, and active transport as previously described. The confinement size L was calculated in short-time delays with 5 s, and the equation to calculate the $L_{confinement}$ is shown below[2]:

$$MSD(t) = A\left(1 - e^{-\frac{1}{\tau}}\right) + 4D_{macro}t + (vt)^2 \tag{2}$$

$$D_{micro} = \frac{A}{4\tau} \tag{3}$$

$$L_{\text{confinement}} = \sqrt{\frac{A}{2}} \qquad (4)$$

## Analysis of telomere puncta intensity

The original data of labeled telomere mean intensity were exported from Fiji with the threshold plugins. We normalized the exported mean intensity and created Fig. 6b by GraphPad Prism 9.

## Statistics and reproducibility

All data generated in this study are provided in the Supplementary Information/Source Data file. The majority of the data were derived from three separate biological replicates. Results are expressed as mean ± standard deviation, with statistical significance defined as a $p$-value $< 0.0001^{****}$.

For Fig. 1e, the Wilcoxon matched-pairs signed rank test was calculated with two-sided p-values using Prism 9 (GraphPad), 69 loci in 23 cells were analyzed, the exact $p$-value is $3.4 \times 10^{-21}$ (left, Fig. 1e). Moreover, the box plot showed the SD, mean, 5%, 95% percentiles, and whiskers, which extend to the minimum and maximum values (right, Fig. 1e). We employed a two-tailed equal-variance Student's $t$-test to compute the $p$-values, and a minimum of three trained independent investigators conducted cell counting in a blinded manner.

For Fig. 4b, curve fittings were performed using OriginPro 9.65, 30 cells were analyzed, and $D_{\text{micro}}$, 95% Confidence band is shown on the top.

## Reporting summary

Further information on research design is available in the Nature Portfolio Reporting Summary linked to this article.

## Data availability

The authors affirm that all data generated or analyzed during this study are accessible through the corresponding author upon request, and source data for the figures and supplementary figures are available as a Source Data file accompanying this publication. There is no restriction to access the source data, and the requests will be fulfilled within two weeks. Source data are provided in this paper.

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

## Acknowledgements

This work was supported by National Science Foundation of China (No. 32271515, 32311530120, X.L. No. 22322409, S.Z.), National Key Research Program (No. 2023YFC2604300, X.L.), the Talent research start-up fund, Beijing Institutes of Life Science, Chinese Academy of Sciences (X.L.), UMass start-up funds (J.W.), and NIH grant R35NS111631 (S.R.J.). We thank Zhenyin Chen, Hongtao Duan, and Jianing Zhong (Gannan Medical University) for their comments and suggestions.

## Author contributions

X.L., Z.Z., and X.R. conceived and designed the experiments. Z.Z., T.X., X.R., and Z.L. contributed plasmids constructions. X.R. and T.X. con-ducted imaging system construction. H.S. contributed to the analysis of Pepper-fused sgRNA stability using RT–qPCR methods. Z.L. and X.R. contributed to the analysis and statistical chromosome heterogeneity data, as well as telomere length. T.X. and Z.Z. contributed to cell line transduction and sorting. H.W., Z.Z., and X.R. conducted live-cell ima-ging genomic DNA double-strand breaks and repair experiments. H.S., H.W., S.Z., J.W., and S.R.J. gave technical support and conceptual advice. X.L., Z.Z., and X.R. wrote the paper with help from all the authors.

## Competing interests

Institute of Zoology, Chinese Academy of Sciences has filed a Chinese patent application (no: 202310878017.6) based on this work. X.L., Z.Z., and X.R. are named as the inventors on the patent application. The remaining authors declare no competing interests.
