## [Peer Review File · Nature Communications]

Fluorogenic CRISPR for genomic DNA imagingReviewer #1 (Remarks to the Author)

Dissecting 3D genome architecture over time (the 4th dimension) is important for us to understand gene regulation and cellular function. CRISPR-based DNA imaging has been recognized as a promising technique to fit this need. Unfortunately, the slow advance of this technology greatly hampers the resolution of the 4th dimension of genome architecture. Thus, developing a versatile, sensitive CRISPR-based DNA imaging system is urgently needed.

Here Zhang et al. develop a method named fCRISPR (fluorogenic CRISPR) for genomic DNA imaging, which substantially increases the signal-to-noise ratio. It is an advance in the CRISPR-based imaging technique. To make the fCRISPR system be utilized by broader users, the authors should provide more quantitative data for labeling sensitivity and specificity. The following are the main comments and suggestions:

In Figure 1, the authors increase the signal-to-noise (S/N) ratio of CRISPR-based DNA imaging by the fCRISPR (tdTomato-tDeg/Pepper-sgRNA) system. To make the comparison more reliable, it will be important to compare the S/N ratio between dCas9-tdTomato/sgRNA and dCas9/Pepper-sgRNA/tdTomato-tDeg under the similar background of tdTomato.

In Figure 2, the authors utilized fCRISPR to detect telomeres, centromeres, C3, and MUC4-I1 in different cell types. All these loci are tandem repeats with high copy numbers. It will be important to demonstrate that fCRISPR could also detect repeats with low copies such as ~20 copies. Statistic data need to be included, especially the percentage of cells with signals, and the average of signal-to-noise ratios.

In Figure 3, the multicolor fCRISPR system (tdTomato-tDeg and Broccoli-BI) should be a very useful tool to detect these loci with low S/N ratios, especially when detecting low-copy repeats. C3 and C13 are tandem repeats with high copy numbers, and it will be important for the authors to detect some loci with ~20 copies, along with some statistical data.

In Figure 4, the heterogeneity of chromosomal dynamics has been shown previously (Chen et al., Cell 2013; Ma et al., JCB 2019, etc). The authors should take advantage of the low background of fCRISPR and investigate whether fCRISPR could be a better tool for the interrogation of chromosomal dynamics. For instance, the authors could compare dCas9-GFP/sgRNA (Chen et al., Cell 2013), CRISPRainbow (Ma et al., NBT 2016), and fCRISPR, to illustrate the difference in chromosomal dynamics when different methods are used.

In Figure 5, CRISPR-based tracking of DSBs and repairs has been done by using LiveFISH (Wang et al., Science 2019). The authors should compare fCRISPR and LiveFISH, to investigate the differences or similarities in the process of DNA damage and repair when different methods are used.

Minor point, page 5, line 201, "the non-repetitive sequences in intron 1 of the MUC4 gene (MUC4-I1) in U2OS cells" is misleading since MUC4-I1 is a tandem repeat in the intron 1 of MUC4, not non-repetitive.

Reviewer #2 (Remarks to the Author)

Review: "Fluorogenic CRISPR for genomic DNA imaging" by Zang Z et al

In this paper, Zhang Z et al. described a novel a system, fCRISPR, for improved imaging of genomic loci in living cells. This system relies on the recruitment of a fluorogenic protein to Pepper inserted into the sgRNA. One the best advantage of this system, is a low background fluorescence and fluorogenic ability, thus increasing the sensitivity of genomic DNA imaging. Indeed, the fluorogenic proteins exhibits fluorescence only when bound to the dCas9-sgRNA complex that is targeted to genomic loci. In this system, the fluorogenic protein is rapidly degraded by the cellular proteasome machinery but become stabilized and fluorescent when they bind the Pepper

aptamer. The advantage of this method, compared to dCas9-GFP, for instance is convincing. The experiments are well done. This method enables genome imaging in different cell lines, and at different genomic loci. In addition, fCRISPR can be used to for multiplexed imaging of different genomic loci when coupled with other CRISPR-based imaging systems.

This article is essentially the description of a new methodology, it is unfortunate that only one application is described.

The proposed application is to detect Double-strand break DNA (DSB) and to visualize their "repair". However, the same type of method, by CRISPR-Cas9-mediated editing, has already been proposed (published) for the detection of DSBs detection and localization of 53BP1. This makes the originality of the this part of the paper questionable.

We can wonder if this reaches the pre-requisites to publish in Nature com.

Concerns:

The author propose one application for this new method: to track Double-strand breaks and analyze/vizualize their repair. This part of the paper should be reinforced, it is not totally convincing:

- There is no indication of the number of cells analyzed.

- The expression profile of 53BP1 (truncated) is strange. Normally, one would expect rather a relatively uniform and diffuse labeling of 53BP1 in the nucleus and not large pre-existing foci/clusters as shown in Figure 5c. (In G1 cells, there may be some nuclear bodies of 53BP1, mainly after replicative stress during the preceding S phase, but this also does not resemble the large foci or aggregate observed in Figure 5c.)

- Couldn't the "resolution" observed (at 3H) come from a change of focal plane? (quite possible with long acquisition times, the cells not being completely frozen or immobile). The pattern of 53BP1 foci or clusters (outside the CBD region) observed in panels 2h and 2h 30 disappears in panel 3H. Video could be shown for this experiment. Different stacks and 3D reconstruction could be applied to support colocalization or not in some experiments (for this one especially).

- To confirm the apparent disappearance of co-localization between the YPet marker (yellow) fused with tDeg (dark red) and the 53BP1-Apple marker (red), authors should consider performing g-H2AX co-labeling (by immunofluorescence after fixation at different times after transfection) and follow the g-H2AX foci co-localization with the marker YPet (yellow) and the tDeg (dark red). The disappearance of H2AX foci should also signal the repair of the DSBs (therefore between 2h and 3H as proposed here).

- The authors observed subsequent dissociation of 53BP1 foci to Chromosome 3 in three hours suggesting that corresponds to repair time of DSB. It will interesting to analyse this "DSB repair" at different genomic loci. Don't you have any observations suggesting repeated cutting and repair of target DNA as well, as observed in the study of Wang et al, *Science* 2019, DOI: 10.1126/science.aax7852 .

- It will be important to compare this timing after genotoxic stress such as ionizing radiation by performing immunofluorescence (g-H2AX foci and 53BP1 foci).

In addition, the authors used a truncated form of 53BP1 (corresponding only to the central region of 53BP1) that is required for its recruitment to chromatin but lacks regions necessary for its regulation (phosphorylation by ATM) or that allows the recruitment of important partners such as RIF1. The authors should use a condition where 53BP1 recruitment should be inhibited to support their data as a negative control.

- The discussion is a little poor, for example the paper of Wang et al could be more evoked, in particular to describe the advantage of this new method for the detection of the DSBs and the

apparent repair of the DSBs.

Minors concerns:

- Details are missing in different figure legends. What type of microscopy used: confocal or epifluorescence? Both are described in the Mat and Method, but it is unclear for which experiments (figures) in particular.

- There seems to be an error in the caption of the figure: "Images were acquired with the Olympus SpinSR-10 microscopy at 3 h time intervals. Image acquisition time, 0.5 h. Scale bar, 5 μm ." It is not 3 hours of intervals, but during 3 hours..

Reviewer #3 (Remarks to the Author)

The authors reported a novel method for visualizing genomic sequences in imaging in living cells based on a CRISPR/Cas9 where a dead Cas9 is combined with Pepper-stabilized fluorogenic protein with a tDeg domain and recognizes sgRNAs that brings also Pepper recognition sequences in their structure.

They present several controls and evidence regarding the the brilliance of the signal, the specificity and versatility of use of their system, benchmarking against the latest similar and used systems (CRISPR-based or MCP based).

Overall, I found this work very solid and well presented. Data are sound and convincing.

I think two important things should be added to their work.

First, if they could show how fCRISPR could work in visualizing real unique sequences, like a promoter or an enhancer region, to convince better the readers of the broadness of using this system compared to others.

How many sgRNAs are needed in these genomic contexts to obtain a convincing signal with the system they are proposing?

Second, it would be worth showing at least once a comparison between the fCRISPR based system and a classic FISH. Indeed, in the FISH approach a BAC is generally used, so a region of at least 60-70 Kb if not larger. With fCRISPR method much smaller regions can be visualized, thus arriving at functional elements of genes for example. The FISH will also prove in reality the specificity of their system.

RESPONSE TO REVIEWERS

We thank the reviewers for their helpful comments and suggestions, and we appreciate the overall enthusiasm they showed for our manuscript. We have performed all the suggested experiments and addressed all the questions raised by the reviewers. There were a few additional experiments requested, which have now been successfully performed. These experiments include:

1. Experiments to compare the signal-to-noise ratio (SNR) between fCRISPR (tdTomato-tDeg reporter) and conventional CRISPR (dCas9-tdTomato reporter) using the same fluorescent protein reporter. To compare SNR between fCRISPR and conventional CRISPR using the same fluorescent protein as reporter, we constructed fCRISPR with tdTomato-tDeg, and conventional CRISPR with dCas9-tdTomato. After transfection and genomic loci imaging, we calculated the SNR of the tdTomato-tDeg reporter and dCas9-tdTomato reporter, respectively. Our new results show that the SNR (means \pm standard deviation) of fCRISPR system (57.48 ± 16.24) is much higher than that of dCas9-tdTomato (1.470 ± 0.2557) with the identical tdTomato reporter. These data are consistent with the results in the original manuscript. These experiments are presented in **Supplementary Figure 4** and **Supplementary Table 1**.
2. Experiments to show fCRISPR can be combined with the Broccoli-fused CRISPR imaging system to detect low-copy (~20 copies) genomic loci across different chromosomes. With fCRISPR, we readily detected low-copy-number (14-30 copies) genomic loci in Chromosome 3, 9, 13, or 19, respectively. With Broccoli-fused CRISPR, we detected genomic loci with low-copy numbers (20-28 copies) in Chromosome 3. In addition, we applied these two orthogonal CRISPR systems to detect two different genomic loci with low-copy numbers simultaneously. Overall, the revised manuscript now demonstrates fCRISPR and the Broccoli-BI CRISPR imaging system can image low-copy (~20 copies) genomic loci in cells. These experiments are presented in **Figure 2b, 3c, Supplementary Figure 8, 11** and **Supplementary Table 3**, and described in the main text.
3. Experiments to compare fCRISPR with other CRISPR-based approaches in the analysis of the chromosomal dynamics heterogeneity. In these experiments, we tracked the dynamics of Chromosome 3 using fCRISPR with tdTomato-tDeg reporter, as well as the other two systems, including conventional CRISPR with dCas9-GFP reporter and MS2-fused CRISPR with MCP-GFP reporter. To characterize chromosomal dynamics, we calculated displacement, and microscopic diffusion coefficients for fCRISPR and other two systems. Based on these calculations, fCRISPR did not show an obvious difference in the analysis of the chromosomal dynamics heterogeneity. However, fCRISPR readily tracks the dynamics of genomic loci due to the low background fluorescence and high SNR compared to the other two systems. These data are included in **Supplementary Figure 12** and **Supplementary Movie 2-4** and described in the main text.
4. Additional data to provide a novel application of fCRISPR in measuring the telomere length. We used fCRISPR to target telomere in retinal pigment epithelium (RPE) cells and human bladder cancer (UMUC3) cells, respectively. fCRISPR shows that telomeres in RPE cells are longer than those in UMUC3 cells. Overall, these results demonstrate that fCRISPR can be used for estimating telomere length. These experiments are presented in **Figure 5** and **Supplementary Figure 13** and described in the main text.

5. Additional data to validate that fCRISPR can detect DNA double-strand breaks (DSBs) and repairs in 53BP1-Apple transduced cells. We used several independent approaches for validation. First, we confirmed that the truncated 53BP1-Apple acts as DSBs sensors by using ATM inhibitor (KU-0055933). Second, we performed immunofluorescence experiments to verify the DSBs event by labeling phosphorylated histone H2AX (γ H2AX). Third, we used three-dimensional reconstruction to precisely image 53BP1 recruitment and resolution. Overall, the revised manuscript now presents multiple approaches to validate the idea that fCRISPR and the 53BP1-Apple reporter are able to track DSBs and repairs. These experiments are presented in **Supplementary Figure 14,15** and described in the main text.
6. Additional data to show more DSBs and repair events. In the revised manuscript we included newly observed DSBs repairing processes by fCRISPR and 53BP1-Apple reporters. First, we observed the repeated DNA cutting and repairing process. Second, we observed homologous-directed repair (HDR) after two homologous Chromosome 3 loci breaks. Third, we demonstrated that fCRISPR can track DSBs and repair in multiple chromosomes. In addition to image *PPP1R2* locus DSBs and repair in Chromosome 3, we observed the DNA breaks and repair of *SPACA7* locus in Chromosome 13 using fCRISPR successfully. These experiments are presented in **Supplementary Figure 17-19 and Supplementary Movie 6-8** and described in the main text.
7. Experiments to confirm the specificity of fCRISPR by fluorescence in situ hybridization (FISH). To do this, we transfected fCRISPR imaging system and incubated FISH probes in U2OS cells. With FISH validation, we confirmed that fCRISPR can specifically detect various loci. The results demonstrate that fCRISPR imaging system is a highly specific technique for genomic labeling and detection. These experiments are presented in **Supplementary Figure 8b,c** and described in the main text.
8. Experiments show that minimal numbers of sgRNAs are needed for genomic loci imaging. We designed fCRISPR with various Pepper-fused sgRNAs targeting genomic loci with different copies. We found that 14 x Pepper-fused sgRNA are needed for genomic loci imaging under confocal imaging. These experiments are presented in **Figure 2b** and **Supplementary Figure 8** and described in the main text.
9. The quantitative analysis on fCRISPR's labeling sensitivity and specificity. In these experiments, we quantified more cells (~50 cells) using fCRISPR in genomic loci imaging than the original manuscript. To analyze fCRISPR's labeling sensitivity, we quantified the SNR of fCRISPR in imaging both high- and low-copy genomic loci. To validate the specificity of fCRISPR, we quantitatively compared fCRISPR to conventional CRISPR and FISH, respectively. Our new quantitative results show high labeling sensitivity and specificity. These experiments are presented in the upper panels of **Figure 1** and **Supplementary Figure 7-9** and are described in the main text.

As a result of these changes, the revised manuscript is substantially improved and now provides more extensive validation of our methodology. We are certain that the manuscript now meets the very high standards required for publication in *Nature Communications*.

We would like to thank the editor and reviewers again for their time and great suggestions to improve our manuscript!

Reviewer #1:

The reviewer says “*Developing a versatile, sensitive CRISPR-based DNA imaging system is urgently needed. fCRISPR is an advance in the CRISPR-based imaging technique.*” The reviewer indicated that the current CRISPR-based DNA imaging systems are greatly hampered in the resolution of the 4th dimension of genome architecture by their slow advance versatility and sensitivity. She/he says, “*Here Zhang et al. develop a method named fCRISPR (fluorogenic CRISPR) for genomic DNA imaging, which substantially increases the signal-to-noise ratio.*” She/he also suggests “*To make the fCRISPR system be utilized by broader users, the authors should provide more quantitative data for labeling sensitivity and specificity.*”

We appreciate the reviewer’s overall positive evaluation that we developed fCRISPR-based DNA imaging system, which substantially increases the signal-to-noise ratio. As the reviewer suggests, we provided comprehensive quantitative data for labeling sensitivity and specificity in the revised manuscript. For details, please see our response to Reviewer 1’s comments below.

1. The reviewer says “*In Figure 1, the authors increase the signal-to-noise (S/N) ratio of CRISPR-based DNA imaging by the fCRISPR (tdTomato-tDeg/Pepper-sgRNA) system. To make the comparison more reliable, it will be important to compare the S/N ratio between dCas9-tdTomato/sgRNA and dCas9/Pepper-sgRNA/tdTomato-tDeg under the similar background of tdTomato.*”

We agree with this important point that the signal-to-noise ratio (SNR) between the conventional CRIPSR system and the fCRISPR system should be compared under the same fluorescent protein reporter of tdTomato. In the original manuscript, we compared the SNR of two systems using different fluorescent proteins. This is because we would like to compare SNR with the identical loci in the same cell using the two systems as shown in Fig. 1b-e in the original manuscript.

We agree with the reviewer’s comments that comparing the conventional CRISPR and fCRISPR under the same reporter will be more reliable. To compare the SNR of two CRISPR systems (conventional CRIPSR with dCas9-tdTomato reporter, fCRISPR with tdTomato-tDeg reporter) with the same tdTomato fluorescent protein, we constructed dCas9-fused tdTomato (dCas9-tdTomato). We expressed dCas9-tdTomato/sgRNA targeting Chromosome 3 in U2OS cells. As a control, we expressed fCRISPR (dCas9/Pepper-sgRNA/tdTomato-tDeg). We found the SNR of fCRISPR (tdTomato-tDeg reporter) is significantly higher than conventional CRIPSR (dCas9-tdTomato reporter) (Supplementary Fig. 4c). Furthermore, we quantified the SNR of fCRISPR and conventional CRIPSR in 22 cells (51 puncta), respectively. The SNR (means \pm standard deviation) of the fCRISPR system (57.48 ± 16.24) is much higher than that of dCas9-tdTomato (1.470 ± 0.2557) using the same tdTomato reporter. In addition, we found that the non-specific aggregation was only observed when using the conventional CRIPSR approach with dCas9-tdTomato, instead of fCRISPR with tdTomato-tDeg (Supplementary Fig. 4a).

The representative images of Chromosome 3 labeled with dCas9-tdTomato and tDeg-tdTomato and SNR data are now included in Supplementary Figure 4, and are described in the main text in the revised manuscript.

2. The reviewer would like to know if fCRISPR can detect repeats with low copies such as ~20 copies in chromosomes, as well as the statistical data including the percentage of cells with signals, and the average of signal-to-noise ratios.

We thank the reviewer for the suggestion. In the original manuscript, we demonstrated that fCRISPR can readily detect the repeat loci with ~90 or higher copy numbers in chromosomes. However, there are very few high-copy (copies>100) chromosome-specific loci in the human genome¹. Therefore, as the reviewer suggests, it is important to image genomic loci with low copies (~20 copies) using fCRISPR.

To test this, we designed sgRNA targeting genomic loci with low copies (~20 copies). Specifically, we first constructed Pepper-fused sgRNAs targeting low-copy genomic loci (5-30 copies)^{1,2}. Next, we co-expressed fCRISPR to detect these low-copy genomic loci in U2OS cells, respectively. Using confocal microscopy, we found that fCRISPR can image low-copy genomic loci varying from 14-30 copies in diverse chromosomes. In addition, we used Fluorescence In Situ Hybridization (FISH) approach for labeling validation and quantification of fCRISPR in 150 cells (Supplementary Fig. 8b,c).

To quantify the average of signal-to-noise ratios, we analyzed the SNR of these fCRISPR labeled low-copy genomic loci with 22 cells. The SNR results of low-copy genomic loci (14-30 copy numbers) labeled by fCRISPR are varying from ~2.2-5.8 (Supplementary Fig. 8d). In contrast, the conventional CRISPR imaging system using a dCas9-fused fluorescent protein reporter barely detect these genomic loci with ~30 copies (Supplementary Fig. 8a), which are consistent with the previous reports²⁻⁴. Thus, this confirms the high sensitivity of fCRISPR for visualizing the low-copy genomic loci.

We additionally quant deletions or duplications. In cancer cells, these mutations will cause them to aneuploidy. We quantified the percentage of cells with signals when observing these low-copy genomic loci in 50 U2OS cancer cell lines. The number of signals is variable in U2OS cells, ~60% of cells showed 2 signals when imaged low-copy numbers in Chromosome 9, 13, and 19, while 46% of cells showed 3 signals in Chromosome 3 (Supplementary Fig. 8e). Thus, fCRISPR can be used to observe the copy number variation by labeling low-copy genomic loci.

Thus, fCRISPR is able to detect low-copy genomic loci with 14-30 copy numbers in chromosomes. These images and quantitative data for specificity, sensitivity, and gene copy number variation are presented in Fig. 2b, Supplementary Fig. 8 and Supplementary Table 3 and are described in the main text.

3. The reviewer says “*In Figure 3, the multicolor fCRISPR system (tdTomato-tDeg and Broccoli-BI) should be a very useful tool to detect these loci with low S/N ratios, especially when detecting low-copy repeats.*” The reviewer suggests that we should use the multicolor orthogonal fCRISPR system to detect low-copy repeats, simultaneously, along with some statistical data.

We agree with the reviewer’s important suggestion. We used the multicolor orthogonal CRISPR systems (fCRISPR with tdTomato-tDeg reporter, and Broccoli-fused CRISPR with BI reporter) to image two different high-copy repeats in the original manuscript. As the reviewer suggests, it would be important to use multicolor orthogonal fCRISPR to detect two different low-copy repeats in the same cell.

As our response to Reviewer 1, point 2, we demonstrated that fCRISPR with tdTomato-tDeg reporter can detect the low-copy genomic loci (~20 copies). We then asked whether Broccoli-fused CRISPR system can detect low-copy genomic loci. To do this, we constructed Broccoli-fused sgRNA targeted various genomic loci with 18-28 copy numbers in Chromosome 3. Next, we co-expressed Broccoli-fused CRISPR to detect these low-copy genomic loci in U2OS cells, respectively. Under confocal microscopy, we found that Broccoli-fused CRISPR can image above 20 copies genomic loci. In addition, we used FISH approach for labeling validation and quantification of Broccoli-fused CRISPR (Supplementary Fig. 11b,c). Therefore, Broccoli-fused CRISPR successfully observed the low-copy genomic loci under confocal microscopy (Supplementary Fig. 11a).

We next constructed the multicolor orthogonal systems with both fCRISPR and Broccoli-fused CRISPR systems, to image two different low-copy genomic loci repeats. After transfection and imaging experiments, we detected the repeats with 14 copies in Chromosome 13 (fCRISPR with tdTomato-tDeg reporter) and 25 copies in Chromosome 3 (Broccoli-fused CRISPR with BI reporter) in the same cell. Therefore, these data showed that the multicolor orthogonal fCRISPR imaging system could detect two different low-copy genomic loci, simultaneously (Fig. 3c).

In the revised manuscript, we now show that Broccoli-fused CRISPR is able to detect low-copy genomic loci with ~20 copies genomic loci. These images and quantitative data for specificity, and sensitivity are presented in Supplementary Fig. 11. Also, the multicolor orthogonal fCRISPR system can be used to image low-copy genomic loci in different chromosomes and are included in Fig. 3c. The targeted sequences and labeling capability are presented in Supplementary Table 3. All these data are described in the main text in the revised manuscript.

4. The reviewer says, “*In Figure 4, the heterogeneity of chromosomal dynamics has been shown previously (Chen et al., Cell 2013; Ma et al., JCB 2019, etc). The authors should take advantage of the low background of fCRISPR and investigate whether fCRISPR could be a better tool for the interrogation of chromosomal dynamics. For instance, the authors could compare dCas9-GFP/sgRNA (Chen et al., Cell 2013), CRISPRainbow (Ma et al., NBT 2016), and fCRISPR, to illustrate the difference in chromosomal dynamics when different methods are used.*”

We thank the reviewer for bringing this point to our attention, and we agree with this valuable point. As the reviewer suggests, we have now compared the dynamics of Chromosome 3 with two other conventional CRISPR systems, including CRISPR with dCas9-GFP reporter (dCas9-GFP/sgRNA)³ and MS2-fused CRISPR with MCP-GFP reporter (CRISPRainbow)⁵. By analyzing the microscopic diffusion coefficients and displacement of labeled Chromosome 3 signals in single cells, we found fCRISPR and other two conventional CRISPR imaging systems showed similar chromosomal dynamics and heterogeneity characteristics (Supplementary Fig. 12a). Furthermore, we compared the confinement of chromosomal dynamics with fCRISPR and these two conventional CRISPR based imaging systems. The quantitative mean-squared displacement (MSD) curves showed that fCRISPR with these two conventional imaging systems had a similar trend with 45 cells, representing similar confinement characteristics (Supplementary Fig. 12b). Therefore, fCRISPR did not show an obvious difference compared to the two other conventional imaging systems in the analysis of the heterogeneity of chromosomal dynamics.

However, as the reviewer points out, fCRISPR exhibits low background and high SNR compared with these two conventional CRISPR systems. The high background of the two other conventional imaging systems likely causes false identification when tracking chromosomal dynamics by using TrackMate plugins in Fiji. Therefore, the chromosomal dynamics tracked by fCRISPR are more readily detected. In addition, the other two conventional CRISPR systems cannot image low-copy genomic loci and track their chromosomal dynamics^{3,5}. Overall, the low background of fCRISPR allows us to readily track the heterogeneity of chromosomal dynamics, and study the dynamics of low-copy genomic loci.

In the revised manuscript, these new data are presented in Supplementary Fig. 12 and Supplementary Movie 2-4.

5. The reviewer points out that we should compare fCRISPR and LiveFISH, to investigate the differences or similarities in the process of DNA damage and repair when used different methods in Figure 5.

We thank the reviewer for the suggestion. In the original manuscript, we transfected fCRISPR imaging system to observe DNA double-strand breaks (DSBs) and repair at *PPP1R2* loci Chromosome 3. In addition, as described in the summary on page 2 (point 6), we performed several additional experiments to study DSBs and repairs. These new data are as follows. First, we observed the repeated DNA cutting and repairing process (Supplementary Fig. 17). Second, we observed homologous-directed repair (HDR) after two homologous Chromosome 3 loci breaks (Supplementary Fig. 18). Third, we demonstrate that fCRISPR can track DSBs and repair in other genomic loci in different chromosomes in addition to *PPP1R2* loci in Chromosome 3 (Supplementary Fig. 19).

We next compared fCRISPR with CRISPR LiveFISH for DNA DSBs and the repair process. Wang et al., Science 2019⁶ reported the CRISPR LiveFISH imaging system to track DSBs repair events. As both fCRISPR and CRISPR LiveFISH used the CRISPR-based imaging approach, most DSB repair events are similar.

First, fCRISPR and CRISPR LiveFISH showed similar repeated cutting and repairing events. For repeated cutting and repairing events, we observed 53BP1 repeatedly recruit and dissociate to Chromosome 3 loci, which represents the *PPP1R2* locus was cutting and repairing multiple times. We have analyzed the time intervals of the second recruiting of 53BP1 foci after the initial dissociation, and the dwell time of 53BP1 in each repair (Supplementary Fig. 17). The time intervals (1-2 h) and dwell time (1-3.5 h) of 53BP1 in repeated cutting and repairing events were observed by fCRISPR. These results are consistent with previous reports of LiveFISH⁶.

Second, fCRISPR and CRISPR LiveFISH showed similar HDR events. For HDR events after DSBs, we observed 53BP1 recruited to two Chromosome 3 loci. After 53BP1 was recruited to both Chromosome 3 loci, these two Chromosome 3 loci (53BP1 existed) gradually got closer and colocalized. After that, the colocalized loci gradually separated away. The 53BP1 foci existed between two separated chromosome loci at last (Supplementary Fig. 18). The dwell time (~3 h) of 53BP1 in the HDR repairing process observed by fCRISPR is similar to CRISPR LiveFISH claimed⁶.

Third, fCRISPR and CRISPR LiveFISH showed similar repair time after DSBs. In the revised manuscript, we quantified the dwell time of 53BP1 foci to Chromosome loci. Most of the 53BP1 foci dissociated with Chromosome loci between 2 to 4 hours after recruitment (60% in 35 cells analyzed) (Supplementary Fig. 16b), which is consistent with CRISPR LiveFISH. In addition, we found a minor difference between fCRISPR and CRISPR LiveFISH. With fCRISPR, we found rapid repairing within 2 h (Fig. 6b,c), which is similar to the previous reports by Liu et al., Science 2020⁷. In contrast, CRISPR LiveFISH did not show the repairing process within as short as 2h.

In summary, fCRISPR and CRISPR LiveFISH mainly observed similar results in tracking DNA DSBs and repair. We now clarify these observations in the revised manuscript.

6. Furthermore, the reviewer points out that “*MUC4-I1* is a tandem repeat in the intron 1 of *MUC4*, not non-repetitive.”

We apologize for any confusion arising from the copy numbers of the intron 1 of *MUC4*. We have corrected the copy numbers of *MUC4-I1* to ~90 repeats in the revised manuscript.

Reviewer #2:

The reviewer says, “*One of the best advantages of this system, is a low background fluorescence and fluorogenic ability, thus increasing the sensitivity of genomic DNA imaging.*” She/he also says, “*The advantage of this method, compared to dCas9-GFP, for instance, is convincing. The experiments are well done. This method enables genome imaging in different cell lines, and at different genomic loci. In addition, fCRISPR can be used for multiplexed imaging of different genomic loci when coupled with other CRISPR-based imaging systems. This article is essentially the description of a new methodology*”. The reviewer suggested that we provide more details of DNA double-strand breaks and more application with fCRISPR.

We appreciate the reviewer’s overall positive evaluation, particularly that our study is essentially the description of a new methodology with low background. We have performed several experiments to enrich DSBs and its repair imaging. In addition, we included a new fCRISPR-based imaging application. For details, please see our response to the below comments.

1. The reviewer suggests that we include another DNA imaging application of fCRISPR in addition to DSBs repair event imaging.

We agree with this important suggestion. fCRISPR imaging approach we described is a new methodology, it would be better to explore its applications more broadly. We have applied fCRISPR to image the DNA breaks and repairs in real-time in the original manuscript.

Based on the reviewer’s comment, we apply fCRISPR to measure the length of telomeres in the revised manuscript. Telomere length is associated with cellular lifespan and tumorigenesis⁸. Telomeres shorten gradually with each cell division and eventually shorten to a certain extent, leading to cell division arrest. Previous studies have found that telomeres in normal human cells are longer than in cancer cell lines³. Therefore, we used fCRISPR to visualize telomeres and to measure the relative telomere length in retinal pigment epithelium (RPE) normal human cells and the human bladder cancer cell line UMUC3.

To do this, we compared the telomere puncta intensity of RPE cells and UMUC3 cells with fCRISPR. The median puncta intensity of telomere in RPE cells was 3.11 times higher than that of telomere in UMUC3 cells (Fig. 5b). This indicates that telomeres in RPE cells are longer than those in UMUC3 cells.

We also compared fCRISPR to conventional CRISPR-based imaging methods in measuring telomere length, and we found that conventional CRISPR-based imaging methods showed 2.85 times higher and matched well with the previous studies³ (Fig. 5b). Therefore, fCRISPR did not show an obvious difference compared to the conventional CRISPR imaging systems. In addition, we used conventional CRISPR system to validate the specificity (Supplementary Fig. 13).

This new application supports that fCRISPR can be used to accurately detect the relative fluorescence brightness of telomeres, thereby inferring telomere length. This further demonstrates that fCRISPR can be used for various DNA imaging applications.

These data are included in Fig. 5 and Supplementary Fig. 13 with statistical analysis in the revised manuscript.

2. The reviewer suggests that we should provide the number of cells analyzed when we present the results of tracking double-strand breaks and analyze/visualize their repair.

We fully agree with this important suggestion. We performed the DSBs experiments and now added statistical data and provided cell numbers in the analysis of DNA DSBs event as below.

We first performed the quantitative analysis for DSBs. We used fCRISPR to image Chromosome 3 and used Cas9 to cut *PPP1R2* in 53BP1-Apple expressing U2OS cells. 12 hours after transfecting cleavage-active Cas9 with *PPP1R2*-targeting sgRNA, we observed that ~308 out of 408 cells (75.5%) showed 53BP1 and fCRISPR colocalization. Without transfection with Cas9 with *PPP1R2*-targeting sgRNA, we observed that only ~13 out of 467 cells (2.8%) showed fCRISPR and 53BP1 colocalization. The cells transfected with scrambled-targeted Cas9 barely showed co-localization, the results showed ~54 out of 478 cells (11.3%) (Supplementary Fig. 14c,d). The quantitative analysis of DSBs demonstrates that fCRISPR can track DSBs in cells, and the formation of 53BP1-Apple foci is highly dependent on Cas9-induced DSBs. These data are shown in Supplementary Fig. 14c,d of the revised manuscript.

We additionally quantified the timing of repair after DNA breaks. We counted the repairing time after DSBs formation in a total of 35 cells. The results showed that 3 out of 35 cells were repaired within 2 h (8.57%), 21 cells were repaired between 2 to 4 h (60.00%) and 11 cells were repaired longer than 4 h (31.43%). Therefore, most cells are repaired between 2 to 4 h after DNA breaks. The quantitative analysis demonstrates that the repairing after Cas9-induced DNA breaks is heterogeneous. These data are included in Supplementary Fig. 16b of the revised manuscript.

In the revised manuscript, we indicated the number of cells in observing DNA DSBs and provided the data in Supplementary Fig. 14c,d and 16b.

3. The reviewer mentioned the expression profile of 53BP1(truncated). She/he says, "*Normally, one would expect rather a relatively uniform and diffuse labeling of 53BP1 in the nucleus and not large pre-existing foci/clusters as shown in Figure 5c.*"

We thank the reviewer for bringing this point to our attention. In the original manuscript, we created the truncated 53BP1-fused Apple transduced U2OS cell line using lentivirus. Since the insertion sites and numbers tend to be heterogeneous, the resulting expression levels of 53BP1-Apple are heterogeneous. The pre-existing foci/clusters are likely due to the overexpression of 53BP1-Apple. To solve this issue, we performed FACS to sort out cells with low expression levels of 53BP1-Apple according to the previously reported approach⁶. In the FACS-sorted cells, we barely observed pre-existing foci/clusters (Supplementary Fig. 14a).

We next tested whether the 53BP1 foci/clusters represent the Cas9-induced DSBs in FACS-sorted cells. To do this, we transfected *PPP1R2*-targeted sgRNA with Cas9 to induce DSBs at a specific locus and empty vector transfection cells as the negative control. We counted the 53BP1-Apple foci/clusters, the results show that after cell sorting, the empty vector-transfected cells show ~4% pre-existing foci/clusters of 53BP1-Apple with ~500 cells analyzed. However, Cas9 with *PPP1R2*-targeted sgRNA transfected cells showed ~92% of fCRISPR loci and 53BP1-Apple foci co-localized with ~500 cells analyzed (Supplementary Fig. 14b). Therefore, the 53BP1 foci/clusters are highly related to Cas9-induced DSBs in the FACS-sorted cells.

In summary, by FACS-sorting cells with lower 53BP1-Apple expression levels, we obtained new stable cell lines in which DNA damage-induced 53BP1 recruitment can be clearly visualized without interference from the pre-existing foci seen in the original cell lines. The use of these re-sorted cell lines with lower background 53BP1-Apple expression allows for visualizing 53BP1 recruitment and dissociation at DNA break sites with high clarity and resolution.

In the revised manuscript, we used newly FACS-sorted U2OS cell lines without pre-existing 53BP1-Apple foci/clusters in all related experiments. Imaging with newly sorted cell lines includes Fig. 6 and Supplementary Fig. 14-19.

4. The reviewer would like to know if “‘resolution’ observed (at 3H) come from a change of focal plane?” She/he says this may be “quite possible with long acquisition times, the cells not being completely frozen or immobile”. Therefore, she/he suggested providing the videos with different stacks and 3D reconstruction to support colocalization between 53BP1 foci and chromosome loci.

We thank the reviewer for bringing this point to our attention. We imaged the DSBs and repair events in live U2OS cell lines, which exhibit flat morphology. Therefore, the ‘resolution’ observed (at 3 h) is unlikely from a change of focal plane in the original manuscript. However, we agree with the reviewer’s point that we should provide the videos with different stacks and 3D reconstruction to support colocalization between 53BP1 foci and chromosome loci.

To determine whether the resolution between 53BP1 and Chromosome 3 is due to the change of focal plane or not, we imaged DSBs events with Z-stacks at different time points. We captured five Z-stack images (~2 μm space) to obtain all the spatial locations of 53BP1-Apple foci around Chromosome 3

loci. The imaging data with different Z-stacks showed that the disappearance of 53BP1-Apple fluorescence was due to the dissociation of 53BP1 from the repair site, rather than the change of focal plane. These data also demonstrate the resolution of 53BP1 at Cas9-induced DNA breaks locus represent the completed repair after DSB, which is consistent with the previous reports^{6,7}.

To present these Z-stack images, we used Imaris x64 (ver. 9.0.1) to create the 3D reconstruction (Supplementary Movies 5-8). In the revised manuscript, we added 3D reconstruction data in Supplementary movies 5-8 which indicates the spatiotemporal relationship between 53BP1 foci and Chromosome 3 loci.

5. The reviewer says “*authors should consider performing γ H2AX co-labeling (by immunofluorescence after fixation at different times after transfection) and follow the γ H2AX foci co-localization with the marker YPet (yellow) and the tDeg (dark red).*” In addition, she/he says “*The disappearance of H2AX foci should also signal the repair of the DSBs*”

We thank the reviewer for bringing this point to our attention. We used multiple biochemical approaches to show that the colocalization between 53BP1-Apple and Chromosome 3 (imaged by fCRISPR) likely suggests DNA breaks. In addition, as described in our response to Reviewer 2, point 4, the resolution between 53BP1-Apple and Chromosome 3 likely suggests the DNA repair. These data and conclusions are consistent with the previous reports^{6,7,9}.

Based on the reviewer’s comments, we tried to perform immunofluorescence of phosphorylated histone H2AX (γ H2AX) with fCRISPR. We found that fCRISPR with YPet-tDeg reporter was fluorescently eliminated after immunolabelling treatment in fixed cells for unknown reasons. Alternatively, we then performed the immunofluorescence of γ H2AX in 53BP1-Apple expressing cells. Several reports indicated that γ H2AX foci appearance can signal the DNA breaks¹⁰⁻¹². We then fixed 53BP1-Apple expressing U2OS cells for γ H2AX immunofluorescence at different time points after the editing-CRISPR transfection for hours. We observed that the γ H2AX and 53BP1 foci colocalization increased after 5 h transfection (~3.5% of cells) until 10 h (~67.7% of cells), each time point with 200 cells analyzed (Supplementary Fig. 15). The γ H2AX and 53BP1 foci colocalization demonstrate Cas9-induced DNA breaks. Therefore, the 53BP1-Apple reporter is the sensor for DNA DSBs, which is consistent with the previous reports^{6,7,9}.

We next asked if the disappearance of γ H2AX signals the repair of the Cas9-induced DSBs. Many reports showed that γ H2AX foci cannot disappear for a long time or exhibit delayed disappearance when Cas9-induced DSBs loci were repaired^{7,13-15}. Therefore, we did not observe the apparent disappearance of γ H2AX and 53BP1 colocalization even after 12 h transfection, which is consistent with the previous report¹³.

To further validate that the disappearance of 53BP1-Apple signals was indeed coming from the repair of DSBs, we then performed the following experiments.

(1). We utilized an ATM inhibitor to demonstrate that the disappearance of 53BP1 signal foci represents the completion of repair. We observed that the addition of an ATM inhibitor prevents the clustered or foci-like structures of 53BP1 reformation at the DNA breaks loci. This finding suggests that the disappearance of 53BP1 at the DNA break loci likely indicates repairing (Supplementary Fig. 14e-i).

(2). We used 3D reconstruction to show the disappearance of 53BP1-Apple after recruitment the repair after DNA breaks. As described in our response to Reviewer 2, point 4, we utilized different Z-stack sections and employed software reconstruction to observe the presence or disappearance of 53BP1 around the chromosomes in cells (Supplementary Movie 5-8).

(3). We performed FACS of stable cell lines expressing 53BP1-Apple to demonstrate the specificity of 53BP1-Apple disappearance after recruitment. As described in our response to Reviewer 2, point 3, the disappearance of 53BP1 foci/clusters in FACS-sorted cells may represent the low expression level, which determines the repairing (Supplementary Fig. 14a-b).

Therefore, it is reasonable to interpret the dissociation of 53BP1-Apple from the targeted chromosomes as the indicator of chromosomal DSBs repair^{6,7}. In the revised manuscript, the immunofluorescence images were shown in Supplementary Fig. 15 and described in the main text.

6. The reviewer says, "*The authors observed subsequent dissociation of 53BP1 foci to Chromosome 3 in three hours suggesting that corresponds to repair time of DSB. It will be interesting to analyze this "DSB repair" at different genomic loci.*"

We agree with this important suggestion. We have previously used fCRISPR to observe the DSBs at *PPP1R2* genomic locus in Chromosome 3. Based on the reviewer's suggestion, we additionally studied DSBs at a different genomic locus in Chromosome 13.

In the revised manuscript, we tracked DSBs and their repair process at *SPACA7* genomic locus in Chromosome 13. For these experiments, we designed fCRISPR system with YPet-tDeg reporter to image Chromosome 13. In addition, we used Cas9 to induce *SPACA7* locus breaks in Chromosome 13. Similar to the method used to observe the *PPP1R2* locus in Chromosome 3, we co-transfected the Chromosome 13 labeled fCRISPR and *SPACA7*-targeted sgRNA with Cas9 to 53BP1-Apple stably expressing U2OS cells. After 7 h of transfection, we observed that 53BP1 gradually recruited to Chromosome 13 locus and dissociate within ~3 hours, likely suggesting the DSBs and repair at *SPACA7* locus in Chromosome 13. These results demonstrate that fCRISPR enables tracking DSBs and repair processes in different chromosomes.

These results are now all included in Supplementary Fig. 19 and Supplementary Movie 8 in the revised manuscript and are described in the main text.

7. The reviewer wonders if the repeated cutting and repair of target DNA is observed.

We thank the reviewer for the constructive comments. Based on the reviewer's comment, we monitored DSBs and repair at *PPP1R2* locus in Chromosome 3.

As we did in the previous manuscript, we co-transfected *PPP1R2*-targeted sgRNA with Cas9 and fCRISPR with YPet-tDeg reporter to 53BP1-Apple expressing U2OS cells. To determine whether fCRISPR can track the repeated cutting and repair at *PPP1R2* locus, we extended the observation time from 3 to 7 h.

As described in the summary on page 2 (point 6) and our response to Reviewer 1, Points 5, we observed repeated recruitment and resolution of 53BP1 foci at Chromosome 3, likely suggesting repeated cutting and repair at the target *PPP1R2* locus. These results are consistent with what Wang et al reported in their paper in *Science*⁶.

These data are shown in Supplementary Fig. 17, and Supplementary Movie 6 in the revised manuscript.

8. The reviewer suggests that we compare the timing of repair after genotoxic stress such as ionizing radiation by performing immunofluorescence.

This is a very interesting question. With fCRISPR-based imaging, we found that the majority of *PPP1R2* locus repair timing after Cas9-induced DSB is between 2 to 4 h (Supplementary Fig. 16b).

It should be noted that the repairing time with ionizing radiation could be longer than Cas9. In addition, ionizing radiation-induced DSBs and repair timing are likely dependent on the dose of irradiation. According to the previous reports¹⁶, when radiation doses are below 1 Gy, the number of γ H2AX and 53BP1 foci reach the maximum ~ 30 minutes after radiation, representing DNA breaks. The γ H2AX and 53BP1 foci gradually decrease representing the completed repair around 2 h¹⁷. When the radiation dose is around 5 Gy, DNA breaks ~30 minutes after radiation, but a significant portion of the foci persist even after 48 h as the DSBs are hard to repair around 5 Gy dose¹². When the radiation doses reach 10 to 20 Gy, γ H2AX or 53BP1 responds more rapidly and forms foci within approximately 15 minutes¹⁸, representing rapid DNA breaking. When the radiation doses exceed 20 Gy, γ H2AX and 53BP1 foci remain highly abundant throughout the observation period¹⁹, indicating that DSBs cannot be repaired at such high radiation doses.

Therefore, the timing of DSBs and repair induced by ionizing radiation or active Cas9 is different. Ionizing radiation likely causes indiscriminate DNA damage. In addition, the DNA damage degree caused by ionizing radiation is higher than that of Cas9.

In the future, we aim to image DSBs repair events when we have the available experimental facilities for studying ionizing radiation. We thank the reviewer for the useful comments. We have now included these differences between Cas9-induced and ionizing radiation-induced DSBs repairs in the discussion in the revised manuscript.

9. The reviewer says, “*In addition, the authors used a truncated form of 53BP1 (corresponding only to the central region of 53BP1) that is required for its recruitment to chromatin but lacks regions necessary for its regulation (phosphorylation by ATM).*” The reviewer suggests that we should “*use a condition where 53BP1 recruitment should be inhibited to support their data as a negative control.*”

We agree with the reviewer that adding a negative control to validate that the truncated 53BP1 is a specific response to DSBs. In the original manuscript, we studied the DNA DSBs and repairs with the truncated version of 53BP1 (amino acid 1220-1711), which has been validated as a well-characterized DNA DSBs sensor in the other reports^{6,7,9}.

Based on the reviewer’s comments, we consider using a condition where 53BP1 recruitment should be inhibited to support our data as a negative control. 53BP1 is required for ataxia telangiectasia-mutated (ATM)-dependent phosphorylation events at sites of DNA breaks¹⁸. Therefore, we used an ATM inhibitor to perform the negative control experiment. To do this, we used ATM inhibitor KU-0055933, a commonly used inhibitor to block ATM-dependent phosphorylation^{20,21}. According to the previous reports, 53BP1 may not recognize and accumulate at DSBs sites after the addition of ATM inhibitor KU-0055933^{7,21}. Therefore, we perform several negative control experiments with ATM inhibitor KU-0055933 as below.

We first asked whether the truncated 53BP1 can recruit to DSBs site in the presence of ATM inhibitor KU-0055933. To do this, we co-transfected the fCRISPR imaging system with YPet-tDeg reporter and *PPP1R2*-targeted Cas9 in 53BP1-Apple expressing U2OS cells, with and without KU-0055933. With KU-0055933, we observed that 53BP1-Apple foci cannot recruit to *PPP1R2* locus. In contrast, 53BP1-Apple foci can easily recruit to *PPP1R2* locus without ATM inhibitor (Supplementary Fig. 14e). We quantified the number of cells with 53BP1 foci and Chromosome 3 loci colocalization. Compared to the cells without KU-0055933, ~76% appeared the co-localization, and the cells with KU-0055933 only had ~5% showing colocalization (Supplementary Fig. 14f). This demonstrates that the truncated form of 53BP1 protein cannot be accumulated at the locus of CRISPR-induced DSB with the addition of ATM inhibitor.

We next asked whether truncated 53BP1, similar to classic biomarker γ H2AX, undergoes phosphorylation following DNA DSBs. Phosphorylated histone H2AX (γ H2AX) functions in the recruitment of DNA damage response proteins to DSBs. ATM also colocalizes with γ H2AX at DSB sites following its auto-phosphorylation. Thus, we tried to observe the localization between 53BP1 and γ H2AX upon the addition of KU-005933. To do this, we co-transfected *PPP1R2*-targeted sgRNA with Cas9 in 53BP1-Apple expressing U2OS cells and performed immunofluorescence to label γ H2AX, with and without the addition of KU-005933. We observed the truncated 53BP1-Apple reporter appeared and colocalized with γ H2AX after Cas9 expression. However, both 53BP1 and γ H2AX formation were abrogated when ATM was suppressed with KU-005933 (Supplementary Fig. 14g). We quantitated the number of cells with 53BP1 and γ H2AX colocalization. The results showed that ~57% of 53BP1 and γ H2AX foci were colocalized without KU-005933, while only ~2.7% colocalization after the addition of KU-005933(Supplementary Fig. 14h).

We also quantified the phosphorylation levels for DSBs-responsive endogenous proteins, including γ H2AX, and pATM. To do this, we transfected *PPP1R2*-targeted sgRNA with Cas9 in 53BP1-Apple expressing U2OS cells and added KU-005933 with different concentrations for Western blotting analysis. Western blots results revealed the reduction of pATM and γ H2AX signal under ATM-inhibited conditions compared to control cells where ATM was active (Supplementary Fig. 14i, 20). These results indicate that the ATM inhibitor effectively inhibits the phosphorylation levels of the DSBs-responsive proteins.

In summary, we have used several approaches to demonstrate that the ATM inhibitor indeed prevented the truncated form of 53BP1 recruitment to the DSBs locus by inhibiting its phosphorylation, consistent with its role as a DNA DSBs sensor^{6,7,9}.

These data are shown in Supplementary Fig. 13e-i, 20with statistical analysis and are described in the revised manuscript.

10. The reviewer says, “*the paper of Wang et al could be more evoked, in particular to describe the advantage of this new method for the detection of the DSBs and the apparent repair of the DSBs*” in the discussion.

We agree with this important suggestion. We included several advantages of fCRISPR in the discussion. For example, we included that fCRISPR shows higher sensitivity to detect the low-copy genomic loci than other conventional systems. We additionally described the advantages of fCRISPR over CRISPR LiveFISH (Wang’s paper⁶) to study DSBs and repair in the discussion in the revised manuscript as below.

“fCRISPR also allows for dynamic tracking of Cas9-induced gene editing and DNA DSBs repair events at endogenous genomic loci in living human cells. Unlike ionizing radiation that induces DSBs in all cellular genomic DNA, Cas9 is able to induce DNA breaks at specific DNA locus with sgRNA. With fCRISPR, we thus spatiotemporally study the Cas9-induced DSB breaks and repair events, including DNA repair timing, for the desired DNA locus.

Like CRISPR LiveFISH, fCRISPR visualizes various DNA breaks and repairing events in living human cells. For example, fCRISPR and CRISPR LiveFISH observed the repeated cutting and repairing, as well as two chromosome loci homologous-directed repairing. However, since fCRISPR is genetically encoded, and does not need to label sgRNA with small molecule fluorophore by the laborious and complicated chemical modification. Therefore, fCRISPR provides a robust and easy-to-operate approach for studying DNA breaks and repair events in living cells with high SNR and low background.”

11. The reviewer suggests we add details figure legends, such as the kind of microscopes for different experiments, respectively.

We apologize for any confusion arising from the type of microscopy. We used Olympus SpinSR10 confocal microscopy in all of the experiments in the revised manuscript. We have changed the description of microscopy in the Methods section.

12. The reviewer points out an error in the caption of the figure: “Images were acquired with the Olympus SpinSR-10 microscopy at 3 h time intervals. Image acquisition time, 0.5 h. Scale bar, 5 μm .” It is not 3 hours of intervals, but during 3 hours.

We apologize for any confusion arising from the time intervals and acquisition time. We have corrected the time of intervals and acquires. We have changed to “Images were acquired with the Olympus SpinSR-10 microscopy at 0.5 h time intervals. Image acquisition time, 3 h. Scale bar, 5 μm .”

Reviewer #3:

The reviewer says, “*The authors reported a novel method for visualizing genomic sequences in imaging in living cells based on a CRISPR/Cas9 where a dead Cas9 is combined with Pepper-stabilized fluorogenic protein with a tDeg domain and recognizes sgRNAs that brings also Pepper recognition sequences in their structure. They present several controls and evidence regarding the brilliance of the signal, the specificity and versatility of use of their system, benchmarking against the latest similar and used systems (CRISPR-based or MCP based).*” She/he also says, “*Overall, I found this work very solid and well presented. Data are sound and convincing.*”

We appreciate the reviewer’s overall positive evaluation, specifically, that this work is very solid and well-presented, and the data are convincing.

1. The reviewer would like to know “*How many sgRNAs are needed in these genomic contexts to obtain a convincing signal with the system they are proposing?*” In addition, the reviewer asked if fCRISPR could work in visualizing unique sequences, like a promoter or an enhancer region.

We thank the reviewer for bringing these points to our attention, we think that these points are very useful to broaden the applications of the fCRISPR system in DNA imaging. In the previous manuscript, we readily imaged *MUC4-11* genomic loci with ~90 repeats (Fig.2b in the original manuscript, Supplementary Fig. 7b in the revised manuscript). In this case, ~90 Pepper-fused sgRNAs are needed to target *MUC4-11* genomic loci.

To determine how many minimal sgRNAs are needed, we designed fCRISPR for targeting genomic loci with 5-30 copies in various chromosomes. After genomic loci imaging under the confocal microscope, we found that fCRISPR can label as low as 14 copy numbers with high sensitivity and specificity (Fig. 2b, Supplementary Fig. 8a-d). In contrast, we barely observed genomic loci with repeats that lower than 14 copy numbers, such as 13 copies genomic loci on Chromosome 9 (Supplementary Table 3). These results indicate that the minimal number of Pepper-fused sgRNAs needed to obtain appearance signals could be 14.

Similarly, 14 unique sgRNAs with different spacers could target the same region for visualizing real unique sequences. However, fCRISPR with many different sgRNAs exhibits several limitations. For example, this approach is difficult to implement for biological applications due to the challenges in the delivery of dozens sgRNAs into the same cells. Furthermore, this approach could increase in off-target sites by the large number of sgRNAs².

However, these limitations could be fixed when we design fCRISPR with single sgRNA fusing multiple Pepper aptamer for signal amplification^{1,2}. This is an ongoing project in the lab.

The images and statistical analysis of fCRISPR labeled 14 copies of genomic loci are shown in Fig. 2b and Supplementary Fig. 8. The targeted sequences and labeling capability are presented in Supplementary Table 3. This section of the content was also described in the main text of the revised manuscript.

2. The reviewer says, “*Second, it would be worth showing at least once a comparison between the fCRISPR based system and a classic FISH. Indeed, in the FISH approach a BAC is generally used, so a region of at least 60-70 Kb if not larger. With fCRISPR method much smaller regions can be visualized, thus arriving at functional elements of genes for example. The FISH will also prove in reality the specificity of their system.*”

We thank the reviewer for the suggestion. In the revised manuscript, we used the classic FISH to verify the specificity of fCRISPR when imaging low-copy genomic loci. We performed FISH verification on fCRISPR labeled low-copy genomic loci in Chromosome 3 (25 copies) and Chromosome 13 (14 copies), respectively. We observed the colocalization between fCRISPR with tdTomato-tDeg reporter and FISH with FITC-fused probes in the same cell (Supplementary Figure 8b). After statistical analysis, there is a 90-92% signal colocalization between fCRISPR and FISH (Supplementary Fig. 8c), representing the high specificity of fCRISPR in low-copy genomic loci imaging.

In addition, we successfully used fCRISPR to label the low-copy genomic loci with 14 copies on Chromosome 13, a region of ~0.5 kb². Thus, as the reviewer points out, fCRISPR could label the smaller region than Bacterial Artificial Chromosome (BAC), which needs around 60-70 kb region.

These data are shown in Fig. 2b and Supplementary Fig. 8 and described in the main text.

We would like to thank all of the reviewers for volunteering their time and effort to provide ideas for enhancing our manuscript. We think that these additional comments and experiments have improved this manuscript and helped to strengthen the conclusions. We hope that with these changes, the reviewers will find the manuscript acceptable for publication in *Nature Communications*.

Reference

- 1 Ma, H. *et al.* CRISPR-Sirius: RNA scaffolds for signal amplification in genome imaging. *Nat. Meth.* **15**, 928-931, doi:10.1038/s41592-018-0174-0 (2018).
- 2 Qin, P. *et al.* Live cell imaging of low- and non-repetitive chromosome loci using CRISPR-Cas9. *Nature Communications* **8**, 14725, doi:10.1038/ncomms14725 (2017).
- 3 Chen, B. *et al.* Dynamic imaging of genomic loci in living human cells by an optimized CRISPR/Cas system. *Cell* **155**, 1479-1491, doi:10.1016/j.cell.2013.12.001 (2013).
- 4 Chen, B. *et al.* Dynamic Imaging of Genomic Loci in Living Human Cells by an Optimized CRISPR/Cas System. *Cell* **155**, 1479-1491, doi:<https://doi.org/10.1016/j.cell.2013.12.001> (2013).
- 5 Ma, H. *et al.* Multiplexed labeling of genomic loci with dCas9 and engineered sgRNAs using CRISPRainbow. *Nature Biotechnology* **34**, 528-530, doi:10.1038/nbt.3526 (2016).
- 6 Wang, H. *et al.* CRISPR-mediated live imaging of genome editing and transcription. *Science* **365**, 1301-1305, doi:doi:10.1126/science.aax7852 (2019).
- 7 Liu, Y. *et al.* Very fast CRISPR on demand. *Science* **368**, 1265-1269, doi:doi:10.1126/science.aay8204 (2020).
- 8 Lansdorp, P. M. *et al.* Heterogeneity in Telomere Length of Human Chromosomes. *Human Molecular Genetics* **5**, 685-691, doi:10.1093/hmg/5.5.685 (1996).
- 9 Yang, K. S., Kohler, R. H., Landon, M., Giedt, R. & Weissleder, R. Single cell resolution in vivo imaging of DNA damage following PARP inhibition. *Scientific Reports* **5**, 10129, doi:10.1038/srep10129 (2015).
- 10 Celeste, A. *et al.* Histone H2AX phosphorylation is dispensable for the initial recognition of DNA breaks. *Nature Cell Biology* **5**, 675-679, doi:10.1038/ncb1004 (2003).
- 11 Kinner, A., Wu, W., Staudt, C. & Iliakis, G. Gamma-H2AX in recognition and signaling of DNA double-strand breaks in the context of chromatin. *Nucleic Acids Res* **36**, 5678-5694, doi:10.1093/nar/gkn550 (2008).
- 12 Redon, C. E., Dickey, J. S., Bonner, W. M. & Sedelnikova, O. A. γ -H2AX as a biomarker of DNA damage induced by ionizing radiation in human peripheral blood lymphocytes and artificial skin. *Advances in Space Research* **43**, 1171-1178, doi:<https://doi.org/10.1016/j.asr.2008.10.011> (2009).
- 13 Brinkman, E. K. *et al.* Quantitative analysis shows that repair of Cas9-induced double-strand DNA breaks is slow and error-prone. *bioRxiv*, 142802, doi:10.1101/142802 (2017).
- 14 Kim, S., Kim, D., Cho, S. W., Kim, J. & Kim, J. S. Highly efficient RNA-guided genome editing in human cells via delivery of purified Cas9 ribonucleoproteins. *Genome Res* **24**, 1012-1019, doi:10.1101/gr.171322.113 (2014).
- 15 Ben-Tov, D. *et al.* Uncovering the Dynamics of Precise Repair at CRISPR/Cas9-induced Double-Strand Breaks. *bioRxiv*, 2023.2001.2010.523377, doi:10.1101/2023.01.10.523377 (2023).
- 16 Vignard, J., Mirey, G. & Salles, B. Ionizing-radiation induced DNA double-strand breaks: A direct and indirect lighting up. *Radiotherapy and Oncology* **108**, 362-369, doi:<https://doi.org/10.1016/j.radonc.2013.06.013> (2013).
- 17 Asaithamby, A. & Chen, D. J. Cellular responses to DNA double-strand breaks after low-dose γ -irradiation. *Nucleic Acids Res.* **37**, 3912-3923, doi:10.1093/nar/gkp237 (2009).
- 18 DiTullio, R. A. *et al.* 53BP1 functions in an ATM-dependent checkpoint pathway that is constitutively activated in human cancer. *Nature Cell Biology* **4**, 998-1002, doi:10.1038/ncb892 (2002).
- 19 BRAVATÀ, V. *et al.* High-dose Ionizing Radiation Regulates Gene Expression Changes in the MCF7 Breast Cancer Cell Line. *Anticancer Research* **35**, 2577-2591 (2015).
- 20 Diehl, M. C., Elmore, L. W. & Holt, S. E. in *Telomeres and Telomerase in Cancer* (ed Keiko Hiyama) 87-125 (Humana Press, 2009).

- 21 Chwastek, J., Jantas, D. & Lasoń, W. The ATM kinase inhibitor KU-55933 provides neuroprotection against hydrogen peroxide-induced cell damage via a γ H2AX/p-p53/caspase-3-independent mechanism: Inhibition of calpain and cathepsin D. *The International Journal of Biochemistry & Cell Biology* **87**, 38-53, doi:<https://doi.org/10.1016/j.biocel.2017.03.015> (2017).

Reviewer #1 (Remarks to the Author)

The Authors have addressed all of my concerns with the original manuscript.

Reviewer #2 (Remarks to the Author)

In this paper, Zhang Z et al. described a novel a system, fCRISPR, for improved imaging of genomic loci in living cells. This system relies on the recruitment of a fluorogenic protein to Pepper inserted into the sgRNA. One the best advantage of this system, is a low background fluorescence and fluorogenic ability, thus increasing the sensitivity of genomic DNA imaging. The advantage of this method, compared to dCas9-GFP, for instance is convincing. The experiments are well done. This method enables genome imaging in different cell lines, and at different genomic loci. In addition, fCRISPR can be used to for multiplexed imaging of different genomic loci when coupled with other CRISPR-based imaging systems.

The authors performed the majority of the suggested experiments and have address now the majority of questions/concerns raised by the reviewers.

We particularly appreciate the authors' effort to:

- include another DNA imaging application of fCRISPR in addition to imaging the DSB repair event. Such as telomere size analysis. They used fCRISPR to visualize telomeres and measure relative telomere length in normal human retinal pigment epithelium (RPE) cells and in the UMUC3 bladder cancer cell line. To do this, they also compared fCRISPR to conventional CRISPR-based imaging methods for measuring telomere length.

- use of re-sorted cell lines with lower background 53BP1-Apple expression allows for visualizing 53BP1 recruitment and dissociation at DNA break sites with high clarity and resolution.

- add imaging data with different Z-stacks showed that the disappearance of 53BP1-Apple fluorescence was due to the dissociation of 53BP1 from the repair site, rather than the change of focal plane.

- Combination with FISH for some experiments

Question:

The authors observed that colocalization of γ H2AX and 53BP1 foci increased from 0 h (~3.5% of 200 cells, 5 h of transfection) to 7 h (~67.7% of 200 cells, 12 h of transfection).

Why does it take so long to observe colocalization between these proteins, since γ -H2AX is also required for 53BP1 recruitment (Supp Fig15), especially since 53BP1 foci are visible as early as 1h (Supp Fig15)?

Comments:

- The authors said « we observed homologous-directed repair (HDR) after two homologous Chromosome 3 loci breaks. »

Be careful not to over-interpret, even if you can see the 2 locis getting closer it is difficult to talk about "HDR repair".

- The authors said "Unlike ionizing radiation that induces DSBs in all cellular genomic DNA, Cas9 is able to induce DNA breaks at specific DNA locus with sgRNA."

Of course, and other Cas9 strategy could be cited that allow targeting DSB breaks with other advantage, for instance the use of light-activated Cas9 to induce breaks and which facilitates nice study on temporal resolution of DSB repair actors (see recent paper of T. Paull, published in Nat Commun, PMID: 37717054).

Reviewer #3 (Remarks to the Author)

The authors have satisfactorily answered to all my previous concerns. I found the new version of the ms greatly improved. I believe this new version is suitable for publication in Nature communication.

RESPONSE TO REVIEWERS

Reviewer #1:

The reviewer says, “*The authors have addressed all of my concerns with the original manuscript.*”

We appreciate the reviewer for her/his satisfaction with the revised manuscript and for comments and evaluation of the manuscript.

Reviewer #2:

1. The reviewer says, “*In this paper, Zhang Z et al. described a **novel** a system, fCRISPR, **for improved imaging of genomic loci in living cells**. This system relies on the recruitment of a fluorogenic protein to Pepper inserted into the sgRNA. One the best advantage of this system, is a low background fluorescence and fluorogenic ability, thus **increasing the sensitivity of genomic DNA imaging**.*”

The advantage of this method, compared to dCas9-GFP, for instance is convincing.

The experiments are well done. This method enables genome imaging in different cell lines, and at different genomic loci. In addition, fCRISPR can be used to for multiplexed imaging of different genomic loci when coupled with other CRISPR-based imaging systems.

The authors performed the majority of the suggested experiments and **have address now the majority of questions/concerns raised by the reviewers.**

We particularly appreciate the authors' effort to:

-include another DNA imaging application of fCRISPR in addition to imaging the DSB repair event. Such as telomere size analysis. They used fCRISPR to visualize telomeres and measure relative telomere length in normal human retinal pigment epithelium (RPE) cells and in the UMUC3 bladder cancer cell line. To do this, they also compared fCRISPR to conventional CRISPR-based imaging methods for measuring telomere length.

- use of re-sorted cell lines with lower background 53BP1-Apple expression allows for visualizing 53BP1 recruitment and dissociation at DNA break sites with high clarity and resolution.

- add imaging data with different Z-stacks showed that the disappearance of 53BP1-Apple fluorescence was due to the dissociation of 53BP1 from the repair site, rather than the change of focal plane.

-Combination with FISH for some experiments”

We thank the reviewer for the overall positive comments, and great suggestions to improve our manuscript. We appreciate that the reviewer satisfies our above effort.

2. The reviewer says, “*The authors observed that colocalization of γ H2AX and 53BP1 foci increased from 0 h (~3.5% of 200 cells, 5 h of transfection) to 7 h (~67.7% of 200 cells, 12 h of transfection).*”

Why does it take so long to observe colocalization between these proteins, since γ -H2AX is also required for 53BP1 recruitment (Supp Fig15), especially since 53BP1 foci are visible as early as 1h (Supp Fig15)?”

We apologize for any confusion about the observation time of colocalization between γ H2AX and 53BP1 foci in Supplementary Fig. 15.

We started to observe colocalization of γ H2AX and 53BP1 after 5 h of transfection in the original Supplementary Fig. 15. Since we used plasmids to express Cas9 and sgRNA, it takes hours for Cas9 and sgRNA to be expressed, and then target and edit the targeted chromosome in cells. In our experiment, we found that Cas9 and sgRNA can be expressed and targeted chromosomes after 5h of transfection, which is consistent with the previous report¹. We therefore started to observe colocalization of γ H2AX and 53BP1 at 0 h (5 h of transfection).

We are sorry we misrepresented the results by using the inappropriate representative figure at 0-2 h (5-7 h of transfection) to show the 53BP1 foci. These 53BP1 foci were barely observed in 0-2h (5-7 h of transfection), and there were 53BP1 foci in the last manuscript which confused the reviewer. Actually, the observed 53BP1-Apple foci were likely naturally occurred and were pre-existing foci/clusters, rather than at the Cas9-induced DSBs site. It should be noted that the pre-existing 53BP1-Apple foci in re-sorted cell lines are very few (23 out of 489 loci, Supplementary Fig. 14a).

Specifically, these existing 53BP1 foci did not colocalize with γ H2AX at 0-2h (5-7 h of transfection), and the statistical data in Supplementary Fig. 15b showed only ~3.5% of cells colocalized. In contrast, we started to observe around ~16.6% of 53BP1 and γ H2AX colocalization at 2 h (7 h of transfection), indicating the onset of the expressed Cas9 starting to induce DSBs.

Therefore, we changed to more appropriate representative figures at 0 h, 1 h, and 2 h to represent the majority of cells in the revised Supplementary Fig. 15a. In the revised figure, we chose the confocal figure without pre-existing 53BP1 foci to avoid misunderstanding.

These updated figures are included in the revised Supplementary Fig. 15a.

3. The reviewer says “*The authors said « we observed homologous-directed repair (HDR) after two homologous Chromosome 3 loci breaks. »*”

Be careful not to over-interpret, even if you can see the 2 locis getting closer it is difficult to talk about "HDR repair".”

We thank the reviewer for bringing this point to our attention, and we agree with this valuable point. We saw two foci getting closer and together after two homologous Chromosome 3 loci breaks. We then interpreted it as the homologous-directed recombination (HDR) event in the last version of manuscript according to the previous references^{2,3}. As the reviewer points out, this may be an over-interpretation as we don't have the direct data to conclude the two-chromosome interaction as HDR after DSBs.

Based on the reviewer's suggestions, we have now interpreted the "two chromosomes loci getting closer and together" event as "possible chromosome interaction" rather than "HDR" in the revised manuscript.

4. The reviewer suggested that we cite papers reporting Cas9 strategies that allow targeting DSB breaks with other advantages. She/he also says, "*for instance the use of light-activated Cas9 to induce breaks and which facilitates nice study on temporal resolution of DSB repair actors (see recent paper of T. Paull, published in Nat Commun, PMID: 37717054).*"

We agree with this important suggestion. We have now cited these papers reporting on light-activated Cas9^{4,5} or sgRNA² strategies that study on temporal resolution of DSB repair events in the revised discussion.

Reviewer #3:

The reviewer says, "*The authors have satisfactorily answered to all my previous concerns. I found the new version of the ms greatly improved. I believe this new version is suitable for publication in Nature Communications.*"

We appreciate the reviewer's satisfaction with the revision. Again, we would like to thank the reviewer for volunteering the time and effort to provide ideas for enhancing our manuscript.

Reference

- 1 Fajrial, A. K., He, Q. Q., Wirusanti, N. I., Slansky, J. E. & Ding, X. A review of emerging physical transfection methods for CRISPR/Cas9-mediated gene editing. *Theranostics* **10**, 5532-5549, doi:10.7150/thno.43465 (2020).
- 2 Liu, Y. *et al.* Very fast CRISPR on demand. *Science* **368**, 1265-1269, doi:doi:10.1126/science.aay8204 (2020).
- 3 Wang, H. *et al.* CRISPR-mediated live imaging of genome editing and transcription. *Science* **365**, 1301-1305, doi:10.1126/science.aax7852 (2019).
- 4 Deshpande, R. A. *et al.* Genome-wide analysis of DNA-PK-bound MRN cleavage products supports a sequential model of DSB repair pathway choice. *Nature Communications* **14**, 5759, doi:10.1038/s41467-023-41544-8 (2023).
- 5 Zou, R. S. *et al.* Massively parallel genomic perturbations with multi-target CRISPR interrogates Cas9 activity and DNA repair at endogenous sites. *Nature Cell Biology* **24**, 1433-1444, doi:10.1038/s41556-022-00975-z (2022).

Reviewer #2 (Remarks to the Author):

The authors have satisfactorily answered to all previous concerns.

I particularly appreciate that the authors answer to my last question about the gH2AX-53BP1 co-localization, and to take account of different suggestions.

I found the ms greatly improved.

This new version is suitable for publication in Nature Communications.